# Towards the renormalisation of the Standard Model effective field theory to dimension eight: Bosonic interactions I

M. Chala[1*], G. Guedes[1,2], M. Ramos[1,2], J. Santiago[1],

**1** CAFPE and Departamento de Física Teórica y del Cosmos, Universidad de Granada, Campus de Fuentenueva, E–18071 Granada, Spain
**2** Laboratório de Instrumentaçao e Física Experimental de Partículas, Departamento de Física da Universidade do Minho, Campus de Gualtar, 4710-057 Braga, Portugal

\* mikael.chala@ugr.es

June 22, 2021

## 1 Abstract

We compute the one-loop renormalisation group running of the bosonic Standard Model effective operators to order $v^4/\Lambda^4$, with $v \sim 246$ GeV being the electroweak scale and $\Lambda$ the unknown new physics threshold. We concentrate on the effects triggered by pairs of the leading dimension-six interactions, namely those that can arise at tree level in weakly-coupled ultraviolet completions of the Standard Model. We highlight some interesting consequences, including the interplay between positivity bounds and the form of the anomalous dimensions; the non renormalisation of the $S$ and $U$ parameters; or the importance of radiative corrections to the Higgs potential for the electroweak phase transition. As a byproduct of this work, we provide a complete Green basis of operators involving only the Higgs and derivatives at dimension-eight, comprising 13 redundant interactions.

## 1   Introduction

The Standard Model (SM) extended with effective interactions, also known as SM effective field theory (SMEFT) [1], is increasingly becoming one of the favourite options for describing particle physics at currently explored energies. The main reasons are the apparent absence of new resonances below the TeV scale [2] and the fact that in general the SMEFT explains the experimental data better than the SM alone [3].

Relatively to the SM, the impact of effective operators of dimension $d > 4$ on observables computed at energy $\sim E$ is of order $(E/\Lambda)^{d-4}$, with $\Lambda \gg E$ being the (unknown) new physics threshold. Thus, the most relevant interactions are those of lowest energy dimension, which, ignoring lepton number violation (LNV), are the ones of dimension six. These operators have been experimentally tested from very different angles at all kind of particle physics facilities. In particular, the knowledge of the corresponding renormalisation group running [4–10] has allowed the high-energy physics community to probe the SMEFT to order $E^2/\Lambda^2$ combining experimental information gathered across very different energies; see for example Refs. [11–14].

However, there is by now convincing evidence that dimension-six operators do not suffice for making predictions within the SMEFT in a number of situations. For example, dimension-six interactions do not provide the dominant contribution to some observables [15] or even they do not arise at tree level in concrete ultraviolet (UV) completions of the SM [16,17]. It can be also that relatively low values of $\Lambda$ are favoured by data in some interactions, and therefore corrections involving higher powers of $E/\Lambda$ are not negligible [16,18]; or simply that some observables are so well measured (or constrained) that they are sensitive to higher-dimensional operators [18–20].

In either case, dimension-eight operators must be retained when using the SMEFT. (Dimension-seven interactions [21,22] are also LNV.) This has been in fact the approach adopted in a number of recent theoretical works [16,18–20,23–28], but so far mostly at tree level. Our goal is to make a first step forward towards the renormalisation of the SMEFT to order $E^4/\Lambda^4$. We think that, beyond opening the door to using the SMEFT precisely and consistently across energy scales, there are several motivations to address this challenge. For example:

**1.** Several classes of dimension-eight operators (including purely bosonic) that arise only at one loop in weakly-coupled UV completions of the SM can be renormalised by dimension-eight terms that can be generated at tree level [29]. (While at dimension six this occurs solely in one case.) This implies that the running of some operators can provide the leading SMEFT corrections to SM predictions in observables in which only loop-induced interactions contribute at tree level.

**2.** Eight is the lowest dimension at which there exist two co-leading contributions to renormalisation within the SMEFT: one involving single insertions of dimension-eight operators, and another one consisting of pairs of dimension-six interactions. (Pairs of dimension-five operators renormalise dimension-six ones [30], but they are LNV and therefore sub-leading with respect to single dimension-six terms.) Non-renormalisation theorems have been established only in relation to the first contribution [17,29]. Thus, whether tree-level dimension-six operators renormalise loop-induced dimension-eight interactions is, to the best of our knowledge, still unknown.

**3.** Dimension-eight operators are subject to positivity bounds [31–37]. Thus, precisely because dimension-six interactions mix into dimension-eight ones, it is *a priori* conceivable that theoretical constraints on combinations of dimension-six Wilson coefficients can be established if the corresponding renormalisation group equations (RGEs) are precisely known.

Inspired by these observations, and in particular by **2**, in this paper we will focus on renormalisation triggered by dimension-six operators. (We will consider the effects of higher-dimensional operators in loops in subsequent works.) Also, we will concentrate on the running of the bosonic sector of the SMEFT.

This article is organised as follows. In section 2 we introduce the relevant Lagrangian and clarify the notation used thorough the rest of the paper. In section 3 we describe the technical details of the renormalisation programme. In section 4 we unravel the global structure of the renormalisation group equations (RGEs). We finalise with a discussion of the results in section 5. We dedicate Appendix A to relations that hold on-shell between different operators. In Appendix B we write explicitly all RGEs, while in Appendix C we describe briefly a UV model that accounts for generic tree-level generated dimension-six bosonic operators.

## 2  Theory and conventions

We denote by $e$, $u$ and $d$ the right-handed (RH) leptons and quarks; while $l$ and $q$ refer to the left-handed (LH) counterparts. The electroweak (EW) gauge bosons and the gluon are named by $W$, $B$ and by $G$, respectively. We represent the Higgs doublet by $\phi = (\phi^+, \phi^0)^T$, and $\tilde{\phi} = \mathrm{i}\sigma_2\phi^*$ with $\sigma_I$ ($I = 1, 2, 3$) being the Pauli matrices. Thus, the renormalisable SM Lagrangian reads:

$$
\begin{aligned}
\mathcal{L}_{\mathrm{SM}} = &-\frac{1}{4}G^A_{\mu\nu}G^{A\,\mu\nu} - \frac{1}{4}W^a_{\mu\nu}W^{a\,\mu\nu} - \frac{1}{4}B_{\mu\nu}B^{\mu\nu} \\
&+ \overline{q^\alpha_L}\mathrm{i}\slashed{D}q^\alpha_L + \overline{l^\alpha_L}\mathrm{i}\slashed{D}l^\alpha_L + \overline{u^\alpha_R}\mathrm{i}\slashed{D}u^\alpha_R + \overline{d^\alpha_R}\mathrm{i}\slashed{D}d^\alpha_R + \overline{e^\alpha_R}\mathrm{i}\slashed{D}e^\alpha_R \\
&+ (D_\mu\phi)^\dagger(D^\mu\phi) - \mu^2|\phi|^2 - \lambda|\phi|^4 - \left(y^u_{\alpha\beta}\overline{q^\alpha_L}\tilde{\phi}u^\beta_R + y^d_{\alpha\beta}\overline{q^\alpha_L}\phi d^\beta_R + y^e_{\alpha\beta}\overline{l^\alpha_L}\phi e^\beta_R + \mathrm{h.c.}\right).
\end{aligned}
\tag{1}
$$

We adopt the minus-sign convention for the covariant derivative:

$$
D_\mu = \partial_\mu - \mathrm{i}g_1 Y B_\mu - \mathrm{i}g_2\frac{\sigma^I}{2}W^I_\mu - \mathrm{i}g_3\frac{\lambda^A}{2}G^A_\mu,
\tag{2}
$$

where $g_1$, $g_2$ and $g_3$ represent, respectively, the $U(1)_Y$, $SU(2)_L$ and $SU(3)_c$ gauge couplings, $Y$ stands for the hypercharge and $\lambda^A$ are the Gell-Mann matrices.

We use the Warsaw basis [38] for the dimension-six SMEFT Lagrangian $\mathcal{L}^{(6)}$, and the basis of Ref. [17] for the dimension-eight part $\mathcal{L}^{(8)}$. (An equivalent basis can be found in Ref. [39].) While the renormalisation of $\mathcal{L}^{(6)}$ has been studied at length [7–9], the running of $\mathcal{L}^{(8)}$ is largely unknown. Assuming lepton-number conservation, the running of dimension-eight Wilson coefficients receives contributions from loops involving single insertions of dimension-eight couplings as well as from pairs of dimension-six operators. Schematically:

$$
16\pi^2\mu\frac{dc^{(8)}_i}{d\mu} = \gamma_{ij}c^{(8)}_j + \gamma'_{ijk}c^{(6)}_j c^{(6)}_k.
\tag{3}
$$

Although $c^{(6)}$ (and $c^{(8)}$) are in general unknown, fits of the SMEFT to the data favour relatively large values of some of these coefficients [3]. This implies that the $\gamma'$ term, which

| | Operator | Notation | Operator | Notation |
|---|---|---|---|---|
| $\phi^8$ | $(\phi^\dagger\phi)^4$ | $\mathcal{O}_{\phi^8}$ | | |
| $\phi^6 D^2$ | $(\phi^\dagger\phi)^2(D_\mu\phi^\dagger D^\mu\phi)$ | $\mathcal{O}_{\phi^6}^{(1)}$ | $(\phi^\dagger\phi)(\phi^\dagger\sigma^I\phi)(D_\mu\phi^\dagger\sigma^I D^\mu\phi)$ | $\mathcal{O}_{\phi^6}^{(2)}$ |
| $\phi^4 D^4$ | $(D_\mu\phi^\dagger D_\nu\phi)(D^\nu\phi^\dagger D^\mu\phi)$ | $\mathcal{O}_{\phi^4}^{(1)}$ | $(D_\mu\phi^\dagger D_\nu\phi)(D^\mu\phi^\dagger D^\nu\phi)$ | $\mathcal{O}_{\phi^4}^{(2)}$ |
| | $(D^\mu\phi^\dagger D_\mu\phi)(D^\nu\phi^\dagger D_\nu\phi)$ | $\mathcal{O}_{\phi^4}^{(3)}$ | | |
| $X^3\phi^2$ | $\epsilon^{IJK}(\phi^\dagger\phi)W_\mu^{I\nu}W_\nu^{J\rho}W_\rho^{K\mu}$ | $\mathcal{O}_{W^3\phi^2}^{(1)}$ | $\epsilon^{IJK}(\phi^\dagger\phi)W_\mu^{I\nu}W_\nu^{J\rho}\widetilde{W}_\rho^{K\mu}$ | $\mathcal{O}_{W^3\phi^2}^{(2)}$ |
| | $\epsilon^{IJK}(\phi^\dagger\sigma^I\phi)B_\mu^{\nu}W_\nu^{J\rho}W_\rho^{K\mu}$ | $\mathcal{O}_{W^2B\phi^2}^{(1)}$ | $\epsilon^{IJK}(\phi^\dagger\sigma^I\phi)(\widetilde{B}^{\mu\nu}W_{\nu\rho}^J W_\mu^{K\rho}+B^{\mu\nu}W_{\nu\rho}^J\widetilde{W}_\mu^{K\rho})$ | $\mathcal{O}_{W^2B\phi^2}^{(2)}$ |
| $X^2\phi^4$ | $(\phi^\dagger\phi)^2 G_{\mu\nu}^A G^{A\mu\nu}$ | $O_{G^2\phi^4}^{(1)}$ | $(\phi^\dagger\phi)^2\widetilde{G}_{\mu\nu}^A G^{A\mu\nu}$ | $O_{G^2\phi^4}^{(2)}$ |
| | $(\phi^\dagger\phi)^2 W_{\mu\nu}^I W^{I\mu\nu}$ | $\mathcal{O}_{W^2\phi^4}^{(1)}$ | $(\phi^\dagger\phi)^2\widetilde{W}_{\mu\nu}^I W^{I\mu\nu}$ | $\mathcal{O}_{W^2\phi^4}^{(2)}$ |
| | $(\phi^\dagger\sigma^I\phi)(\phi^\dagger\sigma^J\phi)W_{\mu\nu}^I W^{J\mu\nu}$ | $\mathcal{O}_{W^2\phi^4}^{(3)}$ | $(\phi^\dagger\sigma^I\phi)(\phi^\dagger\sigma^J\phi)\widetilde{W}_{\mu\nu}^I W^{J\mu\nu}$ | $\mathcal{O}_{W^2\phi^4}^{(4)}$ |
| | $(\phi^\dagger\phi)(\phi^\dagger\sigma^I\phi)W_{\mu\nu}^I B^{\mu\nu}$ | $\mathcal{O}_{WB\phi^4}^{(1)}$ | $(\phi^\dagger\phi)(\phi^\dagger\sigma^I\phi)\widetilde{W}_{\mu\nu}^I B^{\mu\nu}$ | $\mathcal{O}_{WB\phi^4}^{(2)}$ |
| | $(\phi^\dagger\phi)^2 B_{\mu\nu}B^{\mu\nu}$ | $\mathcal{O}_{B^2\phi^4}^{(1)}$ | $(\phi^\dagger\phi)^2\widetilde{B}_{\mu\nu}B^{\mu\nu}$ | $\mathcal{O}_{B^2\phi^4}^{(2)}$ |
| $X^2\phi^2 D^2$ | $(D^\mu\phi^\dagger D^\nu\phi)W_{\mu\rho}^I W_\nu^{I\rho}$ | $\mathcal{O}_{W^2\phi^2 D^2}^{(1)}$ | $(D^\mu\phi^\dagger D_\mu\phi)W_{\nu\rho}^I W^{I\nu\rho}$ | $\mathcal{O}_{W^2\phi^2 D^2}^{(2)}$ |
| | $(D^\mu\phi^\dagger D_\mu\phi)W_{\nu\rho}^I\widetilde{W}^{I\nu\rho}$ | $\mathcal{O}_{W^2\phi^2 D^2}^{(3)}$ | $i\epsilon^{IJK}(D^\mu\phi^\dagger\sigma^I D^\nu\phi)W_{\mu\rho}^J W_\nu^{K\rho}$ | $\mathcal{O}_{W^2\phi^2 D^2}^{(4)}$ |
| | $\epsilon^{IJK}(D^\mu\phi^\dagger\sigma^I D^\nu\phi)(W_{\mu\rho}^J\widetilde{W}_\nu^{K\rho}-\widetilde{W}_{\mu\rho}^J W_\nu^{K\rho})$ | $\mathcal{O}_{W^2\phi^2 D^2}^{(5)}$ | $i\epsilon^{IJK}(D^\mu\phi^\dagger\sigma^I D^\nu\phi)(W_{\mu\rho}^J\widetilde{W}_\nu^{K\rho}+\widetilde{W}_{\mu\rho}^J W_\nu^{K\rho})$ | $\mathcal{O}_{W^2\phi^2 D^2}^{(6)}$ |
| | $(D^\mu\phi^\dagger\sigma^I D_\mu\phi)B_{\nu\rho}W^{I\nu\rho}$ | $\mathcal{O}_{WB\phi^2 D^2}^{(1)}$ | $(D^\mu\phi^\dagger\sigma^I D_\mu\phi)B_{\nu\rho}\widetilde{W}^{I\nu\rho}$ | $\mathcal{O}_{WB\phi^2 D^2}^{(2)}$ |
| | $i(D^\mu\phi^\dagger\sigma^I D^\nu\phi)(B_{\mu\rho}W_\nu^{I\rho}-B_{\nu\rho}W_\mu^{I\rho})$ | $\mathcal{O}_{WB\phi^2 D^2}^{(3)}$ | $(D^\mu\phi^\dagger\sigma^I D^\nu\phi)(B_{\mu\rho}W_\nu^{I\rho}+B_{\nu\rho}W_\mu^{I\rho})$ | $\mathcal{O}_{WB\phi^2 D^2}^{(4)}$ |
| | $i(D^\mu\phi^\dagger\sigma^I D^\nu\phi)(B_{\mu\rho}\widetilde{W}_\nu^{I\rho}-B_{\nu\rho}\widetilde{W}_\mu^{I\rho})$ | $\mathcal{O}_{WB\phi^2 D^2}^{(5)}$ | $(D^\mu\phi^\dagger\sigma^I D^\nu\phi)(B_{\mu\rho}\widetilde{W}_\nu^{I\rho}+B_{\nu\rho}\widetilde{W}_\mu^{I\rho})$ | $\mathcal{O}_{WB\phi^2 D^2}^{(6)}$ |
| | $(D^\mu\phi^\dagger D^\nu\phi)B_{\mu\rho}B_\nu^{\rho}$ | $\mathcal{O}_{B^2\phi^2 D^2}^{(1)}$ | $(D^\mu\phi^\dagger D_\mu\phi)B_{\nu\rho}B^{\nu\rho}$ | $\mathcal{O}_{B^2\phi^2 D^2}^{(2)}$ |
| | $(D^\mu\phi^\dagger D_\mu\phi)B_{\nu\rho}\widetilde{B}^{\nu\rho}$ | $\mathcal{O}_{B^2\phi^2 D^2}^{(3)}$ | | |
| $X\phi^4 D^2$ | $i(\phi^\dagger\phi)(D^\mu\phi^\dagger\sigma^I D^\nu\phi)W_{\mu\nu}^I$ | $\mathcal{O}_{W\phi^4 D^2}^{(1)}$ | $i(\phi^\dagger\phi)(D^\mu\phi^\dagger\sigma^I D^\nu\phi)\widetilde{W}_{\mu\nu}^I$ | $\mathcal{O}_{W\phi^4 D^2}^{(2)}$ |
| | $i\epsilon^{IJK}(\phi^\dagger\sigma^I\phi)(D^\mu\phi^\dagger\sigma^J D^\nu\phi)W_{\mu\nu}^K$ | $\mathcal{O}_{W\phi^4 D^2}^{(3)}$ | $i\epsilon^{IJK}(\phi^\dagger\sigma^I\phi)(D^\mu\phi^\dagger\sigma^J D^\nu\phi)\widetilde{W}_{\mu\nu}^K$ | $\mathcal{O}_{W\phi^4 D^2}^{(4)}$ |
| | $i(\phi^\dagger\phi)(D^\mu\phi^\dagger D^\nu\phi)B_{\mu\nu}$ | $\mathcal{O}_{B\phi^4 D^2}^{(1)}$ | $i(\phi^\dagger\phi)(D^\mu\phi^\dagger D^\nu\phi)\widetilde{B}_{\mu\nu}$ | $\mathcal{O}_{B\phi^4 D^2}^{(2)}$ |

Table 1: *Basis of bosonic dimension-eight operators involving the Higgs. We follow the notation from Ref. [17]. All interactions are hermitian. The operators in grey arise only at one loop in weakly-coupled renormalisable UV completions of the SM [29].*

is quadratic in the dimension-six couplings, can dominate the running of dimension-eight Wilson coefficients even if the latter are equally large. As such, the computation of this piece of the running is especially appealing.

Moreover, non-renormalisation theorems [29, 40–42] have not been yet established for the mixing triggered by pairs of dimension-six operators. Consequently, for now the zeros in $\gamma'$ can be only obtained upon explicit calculation.

We therefore focus on this part of the dimension-eight running in what follows. Likewise, and as a first attack to the problem, we will concentrate on the bosonic sector of the theory. The advantage of this is that bosonic operators are not renormalised by field-redefining away redundant operators involving fermions (the opposite is not true). Besides, we consider loops involving only dimension-six operators that can arise at tree level in weakly-coupled UV completions of the SM. These can be found in Refs. [43–47].

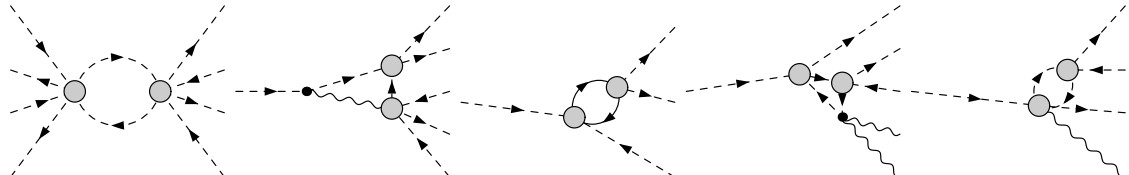

Figure 1: *Example diagrams for the renormalisation of operators in classes $\phi^8$ (first), $\phi^6 D^2$ (second), $\phi^4 D^2$ (third), $X^2 \phi^4$ (fourth) and $X \phi^4 D^2$ (fifth). The gray blobs represent dimension-six interactions.*

Thus, our starting Lagrangian is:

$$
\begin{aligned}
\mathcal{L}_{\mathrm{UV}} = \mathcal{L}_{\mathrm{SM}} + \frac{1}{\Lambda^2} \Big\{ & c_\phi (\phi^\dagger \phi)^3 + c_{\phi\square}(\phi^\dagger \phi)\square(\phi^\dagger \phi) + c_{\phi D}(\phi^\dagger D^\mu \phi)^*(\phi^\dagger D_\mu \phi) \\
& + c^{(1)}_{\phi\psi_L}(\phi^\dagger i \overset{\leftrightarrow}{D}_\mu \phi)(\overline{\psi_L}\gamma^\mu \psi_L) + c^{(3)}_{\phi\psi_L}(\phi^\dagger i \overset{\leftrightarrow}{D}{}^I_\mu \phi)(\overline{\psi_L}\gamma^\mu \sigma^I \psi_L) + c_{\phi\psi_R}(\phi^\dagger i \overset{\leftrightarrow}{D}_\mu \phi)(\overline{\psi_R}\gamma^\mu \psi_R) \\
& + \Big[ c_{\phi ud}(\widetilde{\phi} i D_\mu \phi)(\overline{u_R}\gamma^\mu d_R) + c_{\psi_R \phi}(\phi^\dagger \phi)\overline{\psi_L}\widetilde{\phi}\psi_R + \mathrm{h.c.} \Big] \Big\},
\end{aligned}
\tag{4}
$$

with $\psi_R = u_R, d_R, e_R$ and $\psi_L = q_L, l_L$. Restricting to the bosonic sector of the SMEFT, only dimension-eight operators involving Higgses can be renormalised at one loop from the Lagrangian above. For clarity, we reproduce them in Table 1 following the notation of Ref. [17].

## 3 Computation

We use the background field method and work in the Feynman gauge in dimensional regularisation with space-time dimension $\mathrm{d} = 4 - 2\epsilon$. We compute the one-loop divergences generated by $\mathcal{L}_{\mathrm{UV}}$ using dedicated routines that rely on `FeynRules` [48], `FeynArts` [49] and `FormCalc` [50]. Most of the calculations have been cross-checked using `Match-Maker` [51]. All amplitudes of the kind $X^3\phi^2$ and $X^2\phi^2 D^2$ are finite, hence operators in these classes do not renormalise within our theory.

The bosonic one-loop divergent Lagrangian, involving Higgses, can be written as:

$$
16\pi^2 \epsilon \, \mathcal{L}_{\mathrm{DIV}} = \tilde{K}_\phi (D_\mu \phi)^\dagger (D^\mu \phi) - \tilde{\mu}^2 |\phi|^2 - \tilde{\lambda}|\phi|^4 + \tilde{c}_i^{(6)}\frac{\mathcal{O}_i^{(6)}}{\Lambda^2} + \tilde{c}_j^{(8)}\frac{\mathcal{O}_j^{(8)}}{\Lambda^4},
\tag{5}
$$

where $i$ and $j$ run over elements in the Green bases of operators of dimension-six and dimension-eight, respectively. The former extends the Warsaw basis with the interactions given in Table 2. The bosonic Higgs operators expanding the dimension-eight Green basis and which are redundant in the basis of Table 1 are shown in Table 3. To the best of our knowledge, this last result is completely new.

We are interested in the unknown $\mathcal{O}(E^4/\Lambda^4)$ piece of the renormalisation of bosonic operators. As such, we only provide this new contribution to the aforementioned divergences. The only exception are the Higgs kinetic term and the dimension-six redundant operators, for which we also compute $E^2/\Lambda^2$ corrections, as these generate $E^4/\Lambda^4$ terms when moving to the physical basis by means of field redefinitions. Note also that, since we are dealing with only bosonic operators, we omit flavour indices. Flavourful couplings are written in matrix form (keeping the correct order in the matrix multiplication) and a trace over indices is implicit. We also use the shorthand notation for matrices $A^2 \equiv \mathrm{Tr}\, A^\dagger A$, where $A$ is an arbitrary flavour matrix.

| | Operator | Notation | Operator | Notation |
|---|---|---|---|---|
| $\phi^2 D^4$ | $(D_\mu D^\mu \phi^\dagger)(D_\nu D^\nu \phi)$ | $\mathcal{O}_{D\phi}$ | | |
| $\phi^4 D^2$ | $(\phi^\dagger \phi)(D_\mu \phi)^\dagger (D^\mu \phi)$ | $\mathcal{O}'_{\phi D}$ | $(\phi^\dagger \phi) D^\mu (\phi^\dagger \mathrm{i} \overleftrightarrow{D}_\mu \phi)$ | $\mathcal{O}''_{\phi D}$ |
| $X\phi^2 D^2$ | $D_\nu W^{I\mu\nu}(\phi^\dagger \mathrm{i}\overleftrightarrow{D}^I_\mu \phi)$ | $\mathcal{O}_{WD\phi}$ | $\partial_\nu B^{\mu\nu}(\phi^\dagger \mathrm{i}\overleftrightarrow{D}_\mu \phi)$ | $\mathcal{O}_{BD\phi}$ |

Table 2: *Independent dimension-six bosonic operators involving the Higgs which are redundant with respect to the Warsaw basis. We adopt the notation of Ref. [52].*

| | Operator | Notation | Operator | Notation |
|---|---|---|---|---|
| $\phi^2 D^6$ | $D^2\phi^\dagger D_\mu D_\nu D^\mu D^\nu \phi$ | $\mathcal{O}_{\phi^2}$ | | |
| $\phi^4 D^4$ | $D_\mu \phi^\dagger D^\mu \phi(\phi^\dagger D^2\phi + \mathrm{h.c.})$ | $\mathcal{O}^{(4)}_{\phi^4}$ | $D_\mu \phi^\dagger D^\mu \phi(\phi^\dagger \mathrm{i}D^2\phi + \mathrm{h.c.})$ | $\mathcal{O}^{(5)}_{\phi^4}$ |
| | $(D_\mu \phi^\dagger \phi)(D^2\phi^\dagger D_\mu \phi) + \mathrm{h.c.}$ | $\mathcal{O}^{(6)}_{\phi^4}$ | $(D_\mu \phi^\dagger \phi)(D^2\phi^\dagger \mathrm{i}D_\mu \phi) + \mathrm{h.c.}$ | $\mathcal{O}^{(7)}_{\phi^4}$ |
| | $(D^2\phi^\dagger \phi)(D^2\phi^\dagger \phi) + \mathrm{h.c.}$ | $\mathcal{O}^{(8)}_{\phi^4}$ | $(D^2\phi^\dagger \phi)(\mathrm{i}D^2\phi^\dagger \phi) + \mathrm{h.c.}$ | $\mathcal{O}^{(9)}_{\phi^4}$ |
| | $(D^2\phi^\dagger D^2\phi)(\phi^\dagger \phi)$ | $\mathcal{O}^{(10)}_{\phi^4}$ | $(\phi^\dagger D^2\phi)(D^2\phi^\dagger \phi)$ | $\mathcal{O}^{(11)}_{\phi^4}$ |
| | $(D_\mu \phi^\dagger \phi)(D^\mu \phi^\dagger D^2\phi) + \mathrm{h.c.}$ | $\mathcal{O}^{(12)}_{\phi^4}$ | $(D_\mu \phi^\dagger \phi)(D^\mu \phi^\dagger \mathrm{i}D^2\phi) + \mathrm{h.c.}$ | $\mathcal{O}^{(13)}_{\phi^4}$ |
| $\phi^6 D^2$ | $(\phi^\dagger \phi)^2(\phi^\dagger D^2\phi + \mathrm{h.c.})$ | $\mathcal{O}^{(3)}_{\phi^6}$ | $(\phi^\dagger \phi)^2 D_\mu(\phi^\dagger \mathrm{i}\overleftrightarrow{D}^\mu \phi)$ | $\mathcal{O}^{(4)}_{\phi^6}$ |
| $X\phi^4 D^2$ | $(\phi^\dagger \phi)D_\nu W^{I\mu\nu}(D_\mu \phi^\dagger \sigma^I \phi + \mathrm{h.c.})$ | $\mathcal{O}^{(5)}_{W\phi^4 D^2}$ | $(\phi^\dagger \phi)D_\nu W^{I\mu\nu}(D_\mu \phi^\dagger \mathrm{i}\sigma^I \phi + \mathrm{h.c.})$ | $\mathcal{O}^{(6)}_{W\phi^4 D^2}$ |
| | $\epsilon^{IJK}(D_\mu \phi^\dagger \sigma^I \phi)(\phi^\dagger \sigma^J D_\nu \phi)W^{K\mu\nu}$ | $\mathcal{O}^{(7)}_{W\phi^4 D^2}$ | $(\phi^\dagger \phi)D_\nu B^{\mu\nu}(D_\mu \phi^\dagger \mathrm{i}\phi + \mathrm{h.c.})$ | $\mathcal{O}^{(3)}_{B\phi^4 D^2}$ |

Table 3: *Independent dimension-eight bosonic operators involving the Higgs which are redundant with respect to the basis of Ref. [17]. Redundant operators in the class $X^2\phi^2 D^2$ are not shown. The addition of h.c. when needed implies that all operators are hermitian.*

145 Thus, the divergences of the couplings of dimension $d \leq 4$ read:

$$\tilde{K}_\phi \supset -\frac{1}{2}(c_{\phi D} + 2c_{\phi\Box})\frac{\mu^2}{\Lambda^2}\,, \tag{6}$$

$$\tilde{\lambda} \supset -\frac{3}{2}(c^2_{\phi D} - 4c_{\phi D}c_{\phi\Box} + 8c^2_{\phi\Box})\frac{\mu^4}{\Lambda^4}\,. \tag{7}$$

146 We use the symbol $\supset$ to make explicit that corrections irrelevant for the computation of
147 the $E^4/\Lambda^4$ terms are disregarded.
148 The $\tilde{c}^{(6)}$ couplings in Eq. (5) read:

$$\tilde{c}_{\phi D} \supset c_{\phi D}(5c_{\phi D} - 8c_{\phi\Box})\frac{\mu^2}{\Lambda^2}\,, \tag{8}$$

$$\tilde{c}_{\phi\Box} \supset \frac{1}{4}(c^2_{\phi D} + 24c_{\phi D}c_{\phi\Box} - 48c^2_{\phi\Box})\frac{\mu^2}{\Lambda^2}\,, \tag{9}$$

$$\tilde{c}_\phi \supset 6c_\phi(3c_{\phi D} - 10c_{\phi\Box})\frac{\mu^2}{\Lambda^2} + 12\lambda(c^2_{\phi D} - 6c_{\phi D}c_{\phi\Box} + 12c^2_{\phi\Box})\frac{\mu^2}{\Lambda^2}\,, \tag{10}$$

$$\tilde{c}'_{\phi D} = \frac{1}{2}\left[(3g_2^2 - 3g_1^2 - 4\lambda)c_{\phi D} + (8\lambda - 6g_2^2)c_{\phi\Box}\right]$$

$$- \left[3(c_{u\phi}y^{u\dagger} + y^u c_{u\phi}^\dagger + y^d c_{d\phi}^\dagger + c_{d\phi}y^{d\dagger}) + y^e c_{e\phi}^\dagger + c_{e\phi}y^{e\dagger}\right]$$

$$+ 12c_{\phi q}^{(3)}(y^u y^{u\dagger} + y^d y^{d\dagger}) - 6(c_{\phi ud}y^{d\dagger}y^u + c_{\phi ud}^\dagger y^{u\dagger}y^d) + 4c_{\phi l}^{(3)}y^e y^{e\dagger} \tag{11}$$

$$- (c_{\phi D}^2 + 4c_{\phi D}c_{\phi\Box} - 8c_{\phi\Box}^2)\frac{\mu^2}{\Lambda^2}\,, \tag{12}$$

$$\tilde{c}''_{\phi D} = -\frac{i}{2}\left[3(c_{u\phi}y^{u\dagger} - y^u c_{u\phi}^\dagger + y^d c_{d\phi}^\dagger - c_{d\phi}y^{d\dagger}) + (y^e c_{e\phi}^\dagger - c_{e\phi}y^{e\dagger})\right]\,, \tag{13}$$

$$\tilde{c}_{BD\phi} = -\frac{g_1}{6}\left[c_{\phi D} + c_{\phi\Box} - 4c_{\phi e} - 4c_{\phi l}^{(1)} + 8c_{\phi u} - 4c_{\phi d} + 4c_{\phi q}^{(1)}\right]\,, \tag{14}$$

$$\tilde{c}_{WD\phi} = -\frac{g_2}{6}\left[c_{\phi\Box} + 4c_{\phi l}^{(3)} + 12c_{\phi q}^{(3)}\right]\,; \tag{15}$$

149    while for the $\tilde{c}^{(8)}$ we get:

$$\tilde{c}_{\phi^8} = \frac{3}{16}\left[336c_\phi^2 + (g_1^4 + g_2^4 + 2g_1^2 g_2^2 + 160\lambda^2)c_{\phi D}^2 + 2304\lambda^2 c_{\phi\Box}^2\right.$$

$$+ 512\lambda c_{\phi\Box}c_\phi - 1920\lambda c_{\phi D}c_\phi - 1152\lambda^2 c_{\phi\Box}c_\phi\Big]$$

$$- \left[c_{e\phi}y^{e\dagger}c_{e\phi}y^{e\dagger} + c_{e\phi}^\dagger y^e c_{e\phi}^\dagger y^e + 2c_{e\phi}c_{e\phi}^\dagger y^e y^{e\dagger} + 2c_{e\phi}^\dagger c_{e\phi}y^{e\dagger}y^e\right]$$

$$- 3\left[c_{u\phi}y^{u\dagger}c_{u\phi}y^{u\dagger} + c_{u\phi}^\dagger y^u c_{u\phi}^\dagger y^u + 2c_{u\phi}c_{u\phi}^\dagger y^u y^{u\dagger} + 2c_{u\phi}^\dagger c_{u\phi}y^{u\dagger}y^u\right]$$

$$- 3\left[c_{d\phi}y^{d\dagger}c_{d\phi}y^{d\dagger} + c_{d\phi}^\dagger y^d c_{d\phi}^\dagger y^d + 2c_{d\phi}c_{d\phi}^\dagger y^d y^{d\dagger} + 2c_{d\phi}^\dagger c_{d\phi}y^{d\dagger}y^d\right]\,, \tag{16}$$

$$\tilde{c}_{\phi^4}^{(1)} = \frac{1}{6}\left[11c_{\phi D}^2 - 32c_{\phi D}c_{\phi\Box} + 16c_{\phi\Box}^2 + 24c_{\phi ud}^2 - 24c_{\phi d}^2 - 8c_{\phi e}^2 - 16(c_{\phi l}^{(1)})^2\right.$$

$$\left. + 16(c_{\phi l}^{(3)})^2 - 48(c_{\phi q}^{(1)})^2 + 48(c_{\phi q}^{(3)})^2 - 24c_{\phi u}^2\right]\,, \tag{17}$$

$$\tilde{c}_{\phi^4}^{(2)} = \frac{1}{6}\left[5c_{\phi D}^2 + 16c_{\phi D}c_{\phi\Box} + 16c_{\phi\Box}^2 + 24c_{\phi d}^2 + 8c_{\phi e}^2 + 16(c_{\phi l}^{(1)})^2 + 16(c_{\phi l}^{(3)})^2 + 48(c_{\phi q}^{(1)})^2\right.$$

$$\left. + 48(c_{\phi q}^{(3)})^2 + 24c_{\phi u}^2\right]\,, \tag{18}$$

$$\tilde{c}_{\phi^4}^{(3)} = \frac{1}{6}\left[-7c_{\phi D}^2 + 16c_{\phi D}c_{\phi\Box} + 40c_{\phi\Box}^2 - 32(c_{\phi l}^{(3)})^2 - 96(c_{\phi q}^{(3)})^2 - 24c_{\phi ud}^2\right]\,, \tag{19}$$

$$\tilde{c}_{\phi^4}^{(4)} = \frac{1}{2}\left[-c_{\phi D}^2 - 2c_{\phi D}c_{\phi\Box} + 24c_{\phi\Box}^2\right]\,, \tag{20}$$

$$\tilde{c}_{\phi^4}^{(6)} = c_{\phi D}\left(c_{\phi D} - c_{\phi\Box}\right)\,, \tag{21}$$

$$\tilde{c}_{\phi^4}^{(8)} = \frac{1}{8}\left[c_{\phi D}^2 - 8c_{\phi D}c_{\phi\Box} + 32c_{\phi\Box}^2\right]\,, \tag{22}$$

$$\tilde{c}_{\phi^4}^{(10)} = \frac{1}{2}c_{\phi D}^2\,, \tag{23}$$

$$\tilde{c}_{\phi^4}^{(11)} = \frac{1}{4}\left[c_{\phi D}^2 - 8c_{\phi D}c_{\phi\Box} + 32c_{\phi\Box}^2\right]\,, \tag{24}$$

$$\tilde{c}_{\phi^4}^{(12)} = \frac{1}{2}c_{\phi D}\left(c_{\phi D} + 2c_{\phi\Box}\right)\,, \tag{25}$$

$$\tilde{c}_{\phi^6}^{(1)} = \frac{1}{4}\Bigg[24c_\phi(c_{\phi D} + 8c_{\phi\Box}) - 2c_{\phi D}c_{\phi\Box}(9g_1^2 + 3g_2^2 - 32\lambda) \tag{26}$$

$$+ c_{\phi D}^2(-3g_1^2 - 9g_2^2 + 34\lambda) - 4c_{\phi\Box}^2(9g_1^2 + 15g_2^2 + 112\lambda)\Bigg]$$

$$- 3(c_{d\phi}^2 + c_{u\phi}^2) - c_{e\phi}^2 - (3c_{\phi l}^{(1)} + 5c_{\phi l}^{(3)})\Bigg[y^e c_{e\phi}^\dagger + c_{e\phi}y^{e\dagger}\Bigg]$$

$$+ 9c_{\phi q}^{(1)}\Bigg[y^u c_{u\phi}^\dagger + c_{u\phi}y^{u\dagger} - y^d c_{d\phi}^\dagger - c_{d\phi}y^{d\dagger}\Bigg] - 15c_{\phi q}^{(3)}\Bigg[y^u c_{u\phi}^\dagger + c_{u\phi}y^{u\dagger} + y^d c_{d\phi}^\dagger + c_{d\phi}y^{d\dagger}\Bigg]$$

$$- 3\Bigg[3(c_{\phi u}c_{u\phi}^\dagger y^u + c_{u\phi}c_{\phi u}y^{u\dagger} - c_{d\phi}c_{\phi d}y^{d\dagger} - c_{\phi d}c_{d\phi}^\dagger y^d) - c_{e\phi}c_{\phi e}y^{e\dagger} - c_{\phi e}c_{e\phi}^\dagger y^e\Bigg]$$

$$- 9\Bigg[2(c_{\phi q}^{(1)} - c_{\phi q}^{(3)})y^u c_{\phi u}y^{u\dagger} + 2(c_{\phi q}^{(1)} + c_{\phi q}^{(3)})y^d c_{\phi d}y^{d\dagger}\Bigg] + 9(c_{\phi q}^{(1)}c_{\phi q}^{(3)} + c_{\phi q}^{(3)}c_{\phi q}^{(1)})\Bigg[-y^u y^{u\dagger} + y^d y^{d\dagger}\Bigg]$$

$$+ 15c_{\phi q}^{(3)}\Bigg[y^u y^{u\dagger} + y^d y^{d\dagger}\Bigg]c_{\phi q}^{(3)} + 9c_{\phi q}^{(1)}\Bigg[y^u y^{u\dagger} + y^d y^{d\dagger}\Bigg]c_{\phi q}^{(1)}$$

$$- 3\Bigg[2(c_{\phi l}^{(1)} + c_{\phi l}^{(3)})y^e c_{\phi e}y^{e\dagger} - (c_{\phi l}^{(1)}c_{\phi l}^{(3)} + c_{\phi l}^{(3)}c_{\phi l}^{(1)})y^e y^{e\dagger}\Bigg] + 5c_{\phi l}^{(3)}y^e y^{e\dagger}c_{\phi l}^{(3)} + 3c_{\phi l}^{(1)}y^e y^{e\dagger}c_{\phi l}^{(1)}$$

$$+ \frac{3}{2}\Bigg[c_{\phi ud}c_{\phi ud}^\dagger y^{u\dagger}y^u + c_{\phi ud}^\dagger c_{\phi ud}y^{d\dagger}y^d\Bigg] + 3\Bigg[3(c_{\phi u}y^{u\dagger}y^u c_{\phi u} + c_{\phi d}y^{d\dagger}y^d c_{\phi d}) + c_{\phi e}y^{e\dagger}y^e c_{\phi e}\Bigg]$$

$$- 6c_{\phi q}^{(3)}\Bigg[y^u c_{\phi ud}y^{d\dagger} + y^d c_{\phi ud}^\dagger y^{u\dagger}\Bigg] + 3\Bigg[c_{\phi ud}c_{d\phi}^\dagger y^u + c_{d\phi}c_{\phi ud}^\dagger y^{u\dagger} + c_{u\phi}c_{\phi ud}y^{d\dagger} + y^d c_{\phi ud}^\dagger c_{u\phi}^\dagger\Bigg],$$

$$\tilde{c}_{\phi^6}^{(2)} = \frac{1}{8}\Bigg[72c_\phi c_{\phi D} - 24c_{\phi\Box}(c_{\phi D} + 2c_{\phi\Box})g_1^2 - 3c_{\phi D}(c_{\phi D} + 16c_{\phi\Box})g_2^2 \tag{27}$$

$$+ 32c_{\phi D}(3c_{\phi D} - 8c_{\phi\Box})\lambda\Bigg] - 3\Bigg[c_{\phi ud}c_{d\phi}^\dagger y^u + c_{d\phi}c_{\phi ud}^\dagger y^{u\dagger} + y^d c_{\phi ud}^\dagger c_{u\phi}^\dagger + c_{u\phi}c_{\phi ud}y^{d\dagger}\Bigg]$$

$$- 2\Bigg[c_{\phi l}^{(1)}y^e c_{e\phi}^\dagger + c_{e\phi}y^{e\dagger}c_{\phi l}^{(1)} - 3(c_{\phi q}^{(1)}y^u c_{u\phi}^\dagger + c_{u\phi}y^{u\dagger}c_{\phi q}^{(1)} - c_{\phi q}^{(1)}y^d c_{d\phi}^\dagger - c_{d\phi}y^{d\dagger}c_{\phi q}^{(1)})\Bigg]$$

$$+ 2\Bigg[c_{\phi e}c_{e\phi}^\dagger y^e + c_{e\phi}c_{\phi e}y^{e\dagger} - 3(c_{\phi u}c_{u\phi}^\dagger y^u + c_{u\phi}c_{\phi u}y^{u\dagger} - c_{\phi d}c_{d\phi}^\dagger y^d - c_{d\phi}c_{\phi d}y^{d\dagger})\Bigg]$$

$$- 4\Bigg[c_{\phi l}^{(1)} + c_{\phi l}^{(3)}\Bigg]y^e c_{\phi e}y^{e\dagger} - 12\Bigg[(c_{\phi q}^{(1)} - c_{\phi q}^{(3)})y^u c_{\phi u}y^{u\dagger} + (c_{\phi q}^{(1)} + c_{\phi q}^{(3)})y^d c_{\phi d}y^{d\dagger}\Bigg]$$

$$+ 2\Bigg[c_{\phi l}^{(1)}c_{\phi l}^{(3)} + c_{\phi l}^{(3)}c_{\phi l}^{(1)} + c_{\phi l}^{(1)}c_{\phi l}^{(1)}\Bigg]y^e y^{e\dagger} + 2c_{\phi e}c_{\phi e}y^{e\dagger}y^e - 6\Bigg[c_{\phi q}^{(1)}c_{\phi q}^{(3)} + c_{\phi q}^{(3)}c_{\phi q}^{(1)} - c_{\phi q}^{(1)}c_{\phi q}^{(1)}\Bigg]y^u y^{u\dagger}$$

$$+ 6c_{\phi u}c_{\phi u}y^{u\dagger}y^u + 6\Bigg[c_{\phi q}^{(1)}c_{\phi q}^{(3)} + c_{\phi q}^{(3)}c_{\phi q}^{(1)} + c_{\phi q}^{(1)}c_{\phi q}^{(1)}\Bigg]y^d y^{d\dagger} + 6c_{\phi d}c_{\phi d}y^{d\dagger}y^d$$

$$- \frac{3}{2}\Bigg[c_{\phi ud}c_{\phi ud}^\dagger y^{u\dagger}y^u + c_{\phi ud}^\dagger c_{\phi ud}y^{d\dagger}y^d\Bigg] + 6c_{\phi q}^{(3)}\Bigg[y^d c_{\phi ud}^\dagger y^{u\dagger} + y^u c_{\phi ud}y^{d\dagger}\Bigg],$$

$$\tilde{c}_{\phi^6}^{(3)} = \frac{1}{16}\Bigg[432c_\phi c_{\phi\Box} + (448\lambda - 12g_1^2 - 12g_2^2)c_{\phi D}c_{\phi\Box} - 48c_\phi c_{\phi D} \tag{28}$$

$$- 8(3g_1^2 + 3g_2^2 + 128\lambda)c_{\phi\Box}^2 - (3g_2^2 + 12\lambda)c_{\phi D}^2\Bigg]$$

$$- c_{e\phi}^2 - 3(c_{d\phi}^2 + c_{u\phi}^2) - \frac{1}{2}\Bigg[(c_{\phi l}^{(1)} + c_{\phi l}^{(3)})(y^e c_{e\phi}^\dagger + c_{e\phi}y^{e\dagger}) - c_{\phi e}c_{e\phi}^\dagger y^e - c_{e\phi}c_{\phi e}y^{e\dagger}\Bigg]$$

$$- \Bigg[c_{\phi l}^{(1)} + c_{\phi l}^{(3)}\Bigg]y^e c_{\phi e}y^{e\dagger} + \frac{1}{2}\Bigg[c_{\phi l}^{(1)}c_{\phi l}^{(3)} + c_{\phi l}^{(1)}c_{\phi l}^{(1)} + c_{\phi l}^{(3)}c_{\phi l}^{(1)} + c_{\phi l}^{(3)}c_{\phi l}^{(3)}\Bigg]y^e y^{e\dagger} + \frac{1}{2}c_{\phi e}c_{\phi e}y^{e\dagger}y^e$$

$$- \frac{3}{2}\Bigg[c_{\phi q}^{(1)} + c_{\phi q}^{(3)}\Bigg](y^d c_{d\phi}^\dagger + c_{d\phi}y^{d\dagger}) + \frac{3}{2}\Bigg[c_{\phi d}c_{d\phi}^\dagger y^d + c_{d\phi}c_{\phi d}y^{d\dagger}\Bigg] - 3\Bigg[c_{\phi q}^{(1)} + c_{\phi q}^{(3)}\Bigg]y^d c_{\phi d}y^{d\dagger}$$

$$+ \frac{3}{2} \left[ c_{\phi q}^{(1)} c_{\phi q}^{(3)} + c_{\phi q}^{(1)} c_{\phi q}^{(1)} + c_{\phi q}^{(3)} c_{\phi q}^{(1)} + c_{\phi q}^{(3)} c_{\phi q}^{(3)} \right] y^d y^{d\dagger} + \frac{3}{2} c_{\phi d} c_{\phi d} y^{d\dagger} y^d$$

$$- \frac{3}{2} \left[ (-c_{\phi q}^{(1)} + c_{\phi q}^{(3)})(y^u c_{u\phi}^{\dagger} + c_{u\phi} y^{u\dagger}) + c_{\phi u} c_{u\phi}^{\dagger} y^u + c_{u\phi} c_{\phi u} y^{u\dagger} \right] - 3 \left[ c_{\phi q}^{(1)} - c_{\phi q}^{(3)} \right] y^u c_{u\phi} y^{u\dagger}$$

$$+ \frac{3}{2} \left[ - c_{\phi q}^{(1)} c_{\phi q}^{(3)} + c_{\phi q}^{(1)} c_{\phi q}^{(1)} - c_{\phi q}^{(3)} c_{\phi q}^{(1)} + c_{\phi q}^{(3)} c_{\phi q}^{(3)} \right] y^u y^{u\dagger} + \frac{3}{2} c_{\phi u} c_{\phi u} y^{u\dagger} y^u ,$$

$$\tilde{c}_{\phi^6}^{(4)} = - \frac{3\mathrm{i}}{2} \left[ c_{d\phi}^{\dagger} \left( c_{\phi q}^{(1)} + c_{\phi q}^{(3)} \right) y^d - c_{\phi d} c_{d\phi}^{\dagger} y^d + c_{d\phi} c_{\phi d} y^{d\dagger} - \left( c_{\phi q}^{(1)} + c_{\phi q}^{(3)} \right) c_{d\phi} y^{d\dagger} \right. \tag{29}$$

$$+ c_{u\phi}^{\dagger} \left( c_{\phi q}^{(1)} - c_{\phi q}^{(3)} \right) y^u - c_{\phi u} c_{u\phi}^{\dagger} y^u + c_{u\phi} c_{\phi u} y^{u\dagger} - \left( c_{\phi q}^{(1)} - c_{\phi q}^{(3)} \right) c_{u\phi} y^{u\dagger} \right]$$

$$- \frac{\mathrm{i}}{2} \left[ c_{e\phi}^{\dagger} \left( c_{\phi l}^{(1)} + c_{\phi l}^{(3)} \right) y^e - c_{\phi e} c_{e\phi}^{\dagger} y^e + c_{e\phi} c_{\phi e} y^{e\dagger} - \left( c_{\phi l}^{(1)} + c_{\phi l}^{(3)} \right) c_{e\phi} y^{e\dagger} \right] ,$$

$$\tilde{c}_{W^2\phi^4}^{(1)} = \frac{g_2^2}{12} \left[ c_{\phi D}^2 - 3 c_{\phi D} c_{\phi\Box} - 12(c_{\phi l}^{(3)})^2 - 36(c_{\phi q}^{(3)})^2 + 9 c_{\phi ud}^2 \right] , \tag{30}$$

$$\tilde{c}_{W^2\phi^4}^{(3)} = \frac{g_2^2}{48} \left[ c_{\phi D}^2 - 12 c_{\phi D} c_{\phi\Box} - 24 c_{\phi d}^2 - 8 c_{\phi e}^2 - 16(c_{\phi l}^{(1)})^2 - 48(c_{\phi q}^{(1)})^2 - 24 c_{\phi u}^2 + 12 c_{\phi ud}^2 \right] , \tag{31}$$

$$\tilde{c}_{WB\phi^4}^{(1)} = \frac{g_1 g_2}{24} \left[ c_{\phi D}^2 - 12 c_{\phi D} c_{\phi\Box} - 8 c_{\phi e}^2 - 16(c_{\phi l}^{(1)})^2 - 48(c_{\phi q}^{(1)})^2 - 24 c_{\phi u}^2 + 12 c_{\phi ud} - 24 c_{\phi d}^2 \right] , \tag{32}$$

$$\tilde{c}_{B^2\phi^4}^{(1)} = - \frac{g_1^2}{48} \left[ 3 c_{\phi D}^2 + 24 c_{\phi d}^2 + 8 c_{\phi e}^2 + 16(c_{\phi l}^{(1)})^2 - 48(c_{\phi l}^{(3)})^2 + 48(c_{\phi q}^{(1)})^2 \right.$$

$$\left. - 144(c_{\phi q}^{(3)})^2 + 24 c_{\phi u}^2 + 24 c_{\phi ud}^2 \right] , \tag{33}$$

$$\tilde{c}_{W\phi^4 D^2}^{(1)} = \frac{g_2}{6} \left[ 5 c_{\phi D}^2 - 24 c_{\phi D} c_{\phi\Box} - 8 c_{\phi e}^2 - 16(c_{\phi l}^{(1)})^2 - 48(c_{\phi l}^{(3)})^2 - 48(c_{\phi q}^{(1)})^2 \right.$$

$$\left. - 144(c_{\phi q}^{(3)})^2 - 24 c_{\phi u}^2 + 48 c_{\phi ud}^2 - 24 c_{\phi d}^2 \right] , \tag{34}$$

$$\tilde{c}_{B\phi^4 D^2}^{(1)} = - \frac{g_1}{6} \left[ 24 c_{\phi d}^2 + 3 c_{\phi D}^2 + 8(c_{\phi e}^2 + 2(c_{\phi l}^{(1)})^2 - 6(c_{\phi l}^{(3)})^2 + 6(c_{\phi q}^{(1)})^2 - 18(c_{\phi q}^{(3)})^2 + 3 c_{\phi u}^2) + 24 c_{\phi ud}^2 \right] , \tag{35}$$

$$\tilde{c}_{W\phi^4 D^2}^{(6)} = \frac{g_2}{24} \left[ 5 c_{\phi D}^2 - 20 c_{\phi D} c_{\phi\Box} + 16(c_{\phi\Box}^2 - 2(c_{\phi l}^{(3)})^2 - 6(c_{\phi q}^{(3)})^2) + 48 c_{\phi ud}^2 \right] , \tag{36}$$

$$\tilde{c}_{W\phi^4 D^2}^{(7)} = \frac{g_2}{6} \left[ 24 c_{\phi d}^2 - c_{\phi D}(c_{\phi D} - 12 c_{\phi\Box}) + 8(c_{\phi e}^2 + 2(c_{\phi l}^{(1)})^2 + 6(c_{\phi q}^{(1)})^2 + 3 c_{\phi u}^2) - 12 c_{\phi ud}^2 \right] , \tag{37}$$

$$\tilde{c}_{B\phi^4 D^2}^{(3)} = \frac{g_1}{24} \left[ - 3 c_{\phi D}^2 + 4 c_{\phi D} c_{\phi\Box} + 16(c_{\phi\Box}^2 + 4(c_{\phi l}^{(3)})^2 + 12(c_{\phi q}^{(3)})^2) - 24 c_{\phi ud}^2 \right] . \tag{38}$$

All other relevant counterterms ($\tilde{\mu}^2$, $\tilde{c}_{D\phi}$, $\tilde{c}_{\phi^4}^{(5,7,9,13)}$, etc.) vanish at the order of $E/\Lambda$ we are interested in.

# 4   The structure of the renormalisation group equations

The explicit form of the divergences in the Green basis, shown in the equations above, is of utmost importance, or else the computation of RGEs at higher orders or involving other light degrees of freedom could not be built on our results [53]. Subsequently, though, one

156 can reduce the redundant operators to the physical basis. We do that following the results
157 in Appendix A.

158    In the physical basis, the RGEs of the different dimension-eight couplings read:

$$16\pi^2\mu\frac{dc_i^{(8)}}{d\mu} = -c_i^{(8)}\sum_j x_j n_j\frac{\partial}{\partial x_j}\frac{\tilde{c}_i^{(8)}}{c_i^{(8)}}\,, \tag{39}$$

159 with $x$ running over all couplings, renormalisable or not, and with $n$ representing the
160 corresponding tree-level anomalous dimension, defined as the value required to keep the
161 couplings dimensionless in $4-2\epsilon$ dimensions. The minus sign results from requiring that
162 the counterterms cancel the divergences. The complete set of RGEs can be found in
163 Appendix B, including those of renormalisable and dimension-six terms. In this section,
164 we limit ourselves to discussing the structure of the $\gamma'$ matrices defined in Eq. (3).

165    Since the contribution comes from pairs of dimension-six operators, we provide a (sym-
166 metric) matrix for each $c_i^{(8)}$ in which we represent with a × a non-vanishing entry, with 0
167 a trivial zero, for which all contributions in the Green basis vanish (for example in some
168 cases there are no diagrams contributing to the corresponding amplitude), and with $\emptyset$ a
169 non-trivial zero, for which several non-vanishing contributions cancel in the physical basis.
170 We find:

171

**$\gamma'_{\mathbf{c}_\phi^{\mathbf{8}}}$**

| | $c_\phi$ | $c_{\phi D}$ | $c_{\phi\Box}$ | $c_{\phi\psi_L}^{(1)}$ | $c_{\phi\psi_L}^{(3)}$ | $c_{\phi\psi_R}$ | $c_{\phi ud}$ | $c_{\psi_R\phi}$ |
|---|---|---|---|---|---|---|---|---|
| $c_\phi$ | × | × | × | 0 | × | 0 | × | × |
| $c_{\phi D}$ | | × | × | × | × | × | × | × |
| $c_{\phi\Box}$ | | | × | 0 | × | 0 | × | × |
| $c_{\phi\psi_L}^{(1)}$ | | | | × | × | × | 0 | × |
| $c_{\phi\psi_L}^{(3)}$ | | | | | × | × | 0 | × |
| $c_{\phi\psi_R}$ | | | | | | × | 0 | × |
| $c_{\phi ud}$ | | | | | | | × | 0 |
| $c_{\psi_R\phi}$ | | | | | | | | × |

**$\gamma'_{\mathbf{c}_{\phi^4}^{(1)}}$**

| | $c_\phi$ | $c_{\phi D}$ | $c_{\phi\Box}$ | $c_{\phi\psi_L}^{(1)}$ | $c_{\phi\psi_L}^{(3)}$ | $c_{\phi\psi_R}$ | $c_{\phi ud}$ | $c_{\psi_R\phi}$ |
|---|---|---|---|---|---|---|---|---|
| $c_\phi$ | 0 | 0 | 0 | 0 | 0 | 0 | 0 | 0 |
| $c_{\phi D}$ | | × | × | 0 | 0 | 0 | 0 | 0 |
| $c_{\phi\Box}$ | | | × | 0 | 0 | 0 | 0 | 0 |
| $c_{\phi\psi_L}^{(1)}$ | | | | × | 0 | 0 | 0 | 0 |
| $c_{\phi\psi_L}^{(3)}$ | | | | | × | 0 | 0 | 0 |
| $c_{\phi\psi_R}$ | | | | | | × | 0 | 0 |
| $c_{\phi ud}$ | | | | | | | × | 0 |
| $c_{\psi_R\phi}$ | | | | | | | | 0 |

172

**$\gamma'_{\mathbf{c}_{\phi^4}^{(2)}}$**

| | $c_\phi$ | $c_{\phi D}$ | $c_{\phi\Box}$ | $c_{\phi\psi_L}^{(1)}$ | $c_{\phi\psi_L}^{(3)}$ | $c_{\phi\psi_R}$ | $c_{\phi ud}$ | $c_{\psi_R\phi}$ |
|---|---|---|---|---|---|---|---|---|
| $c_\phi$ | 0 | 0 | 0 | 0 | 0 | 0 | 0 | 0 |
| $c_{\phi D}$ | | × | × | 0 | 0 | 0 | 0 | 0 |
| $c_{\phi\Box}$ | | | × | 0 | 0 | 0 | 0 | 0 |
| $c_{\phi\psi_L}^{(1)}$ | | | | × | 0 | 0 | 0 | 0 |
| $c_{\phi\psi_L}^{(3)}$ | | | | | × | 0 | 0 | 0 |
| $c_{\phi\psi_R}$ | | | | | | × | 0 | 0 |
| $c_{\phi ud}$ | | | | | | | 0 | 0 |
| $c_{\psi_R\phi}$ | | | | | | | | 0 |

**$\gamma'_{\mathbf{c}_{\phi^4}^{(3)}}$**

| | $c_\phi$ | $c_{\phi D}$ | $c_{\phi\Box}$ | $c_{\phi\psi_L}^{(1)}$ | $c_{\phi\psi_L}^{(3)}$ | $c_{\phi\psi_R}$ | $c_{\phi ud}$ | $c_{\psi_R\phi}$ |
|---|---|---|---|---|---|---|---|---|
| $c_\phi$ | 0 | 0 | 0 | 0 | 0 | 0 | 0 | 0 |
| $c_{\phi D}$ | | × | × | 0 | 0 | 0 | 0 | 0 |
| $c_{\phi\Box}$ | | | × | 0 | 0 | 0 | 0 | 0 |
| $c_{\phi\psi_L}^{(1)}$ | | | | 0 | 0 | 0 | 0 | 0 |
| $c_{\phi\psi_L}^{(3)}$ | | | | | × | 0 | 0 | 0 |
| $c_{\phi\psi_R}$ | | | | | | 0 | 0 | 0 |
| $c_{\phi ud}$ | | | | | | | × | 0 |
| $c_{\psi_R\phi}$ | | | | | | | | 0 |

173

**$\gamma'_{\mathbf{c}_{\phi^6}^{(1)}}$**

| | $c_\phi$ | $c_{\phi D}$ | $c_{\phi\Box}$ | $c_{\phi\psi_L}^{(1)}$ | $c_{\phi\psi_L}^{(3)}$ | $c_{\phi\psi_R}$ | $c_{\phi ud}$ | $c_{\psi_R\phi}$ |
|---|---|---|---|---|---|---|---|---|
| $c_\phi$ | 0 | × | × | 0 | 0 | 0 | 0 | 0 |
| $c_{\phi D}$ | | × | × | × | × | × | × | × |
| $c_{\phi\Box}$ | | | × | 0 | × | 0 | × | × |
| $c_{\phi\psi_L}^{(1)}$ | | | | × | × | × | 0 | × |
| $c_{\phi\psi_L}^{(3)}$ | | | | | × | × | × | × |
| $c_{\phi\psi_R}$ | | | | | | × | 0 | × |
| $c_{\phi ud}$ | | | | | | | × | × |
| $c_{\psi_R\phi}$ | | | | | | | | × |

**$\gamma'_{\mathbf{c}_{\phi^6}^{(2)}}$**

| | $c_\phi$ | $c_{\phi D}$ | $c_{\phi\Box}$ | $c_{\phi\psi_L}^{(1)}$ | $c_{\phi\psi_L}^{(3)}$ | $c_{\phi\psi_R}$ | $c_{\phi ud}$ | $c_{\psi_R\phi}$ |
|---|---|---|---|---|---|---|---|---|
| $c_\phi$ | 0 | × | 0 | 0 | 0 | 0 | 0 | 0 |
| $c_{\phi D}$ | | × | × | × | × | × | × | × |
| $c_{\phi\Box}$ | | | × | 0 | 0 | 0 | 0 | 0 |
| $c_{\phi\psi_L}^{(1)}$ | | | | × | × | × | 0 | × |
| $c_{\phi\psi_L}^{(3)}$ | | | | | × | × | × | 0 |
| $c_{\phi\psi_R}$ | | | | | | × | 0 | × |
| $c_{\phi ud}$ | | | | | | | × | × |
| $c_{\psi_R\phi}$ | | | | | | | | × |

174

| $\gamma'_{c^{(1)}_{W^2\phi^4}}$ | $c_\phi$ | $c_{\phi D}$ | $c_{\phi\square}$ | $c^{(1)}_{\phi\psi_L}$ | $c^{(3)}_{\phi\psi_L}$ | $c_{\phi\psi_R}$ | $c_{\phi ud}$ | $c_{\psi_R\phi}$ |
|---|---|---|---|---|---|---|---|---|
| $c_\phi$ | 0 | 0 | 0 | 0 | 0 | 0 | 0 | 0 |
| $c_{\phi D}$ | | × | ∅ | 0 | 0 | 0 | 0 | 0 |
| $c_{\phi\square}$ | | | 0 | 0 | 0 | 0 | 0 | 0 |
| $c^{(1)}_{\phi\psi_L}$ | | | | × | 0 | 0 | 0 | 0 |
| $c^{(3)}_{\phi\psi_L}$ | | | | | × | 0 | 0 | 0 |
| $c_{\phi\psi_R}$ | | | | | | × | 0 | 0 |
| $c_{\phi ud}$ | | | | | | | × | 0 |
| $c_{\psi_R\phi}$ | | | | | | | | 0 |

| $\gamma'_{c^{(3)}_{W^2\phi^4}}$ | $c_\phi$ | $c_{\phi D}$ | $c_{\phi\square}$ | $c^{(1)}_{\phi\psi_L}$ | $c^{(3)}_{\phi\psi_L}$ | $c_{\phi\psi_R}$ | $c_{\phi ud}$ | $c_{\psi_R\phi}$ |
|---|---|---|---|---|---|---|---|---|
| $c_\phi$ | 0 | 0 | 0 | 0 | 0 | 0 | 0 | 0 |
| $c_{\phi D}$ | | ∅ | ∅ | 0 | 0 | 0 | 0 | 0 |
| $c_{\phi\square}$ | | | 0 | 0 | 0 | 0 | 0 | 0 |
| $c^{(1)}_{\phi\psi_L}$ | | | | ∅ | 0 | 0 | 0 | 0 |
| $c^{(3)}_{\phi\psi_L}$ | | | | | 0 | 0 | 0 | 0 |
| $c_{\phi\psi_R}$ | | | | | | ∅ | 0 | 0 |
| $c_{\phi ud}$ | | | | | | | ∅ | 0 |
| $c_{\psi_R\phi}$ | | | | | | | | 0 |

175

| $\gamma'_{c^{(1)}_{WB\phi^4}}$ | $c_\phi$ | $c_{\phi D}$ | $c_{\phi\square}$ | $c^{(1)}_{\phi\psi_L}$ | $c^{(3)}_{\phi\psi_L}$ | $c_{\phi\psi_R}$ | $c_{\phi ud}$ | $c_{\psi_R\phi}$ |
|---|---|---|---|---|---|---|---|---|
| $c_\phi$ | 0 | 0 | 0 | 0 | 0 | 0 | 0 | 0 |
| $c_{\phi D}$ | | ∅ | ∅ | 0 | 0 | 0 | 0 | 0 |
| $c_{\phi\square}$ | | | 0 | 0 | 0 | 0 | 0 | 0 |
| $c^{(1)}_{\phi\psi_L}$ | | | | ∅ | 0 | 0 | 0 | 0 |
| $c^{(3)}_{\phi\psi_L}$ | | | | | 0 | 0 | 0 | 0 |
| $c_{\phi\psi_R}$ | | | | | | ∅ | 0 | 0 |
| $c_{\phi ud}$ | | | | | | | ∅ | 0 |
| $c_{\psi_R\phi}$ | | | | | | | | 0 |

| $\gamma'_{c^{(1)}_{B^2\phi^4}}$ | $c_\phi$ | $c_{\phi D}$ | $c_{\phi\square}$ | $c^{(1)}_{\phi\psi_L}$ | $c^{(3)}_{\phi\psi_L}$ | $c_{\phi\psi_R}$ | $c_{\phi ud}$ | $c_{\psi_R\phi}$ |
|---|---|---|---|---|---|---|---|---|
| $c_\phi$ | 0 | 0 | 0 | 0 | 0 | 0 | 0 | 0 |
| $c_{\phi D}$ | | × | 0 | 0 | 0 | 0 | 0 | 0 |
| $c_{\phi\square}$ | | | 0 | 0 | 0 | 0 | 0 | 0 |
| $c^{(1)}_{\phi\psi_L}$ | | | | × | 0 | 0 | 0 | 0 |
| $c^{(3)}_{\phi\psi_L}$ | | | | | × | 0 | 0 | 0 |
| $c_{\phi\psi_R}$ | | | | | | × | 0 | 0 |
| $c_{\phi ud}$ | | | | | | | × | 0 |
| $c_{\psi_R\phi}$ | | | | | | | | 0 |

176

| $\gamma'_{c^{(1)}_{W\phi^4 D^2}}$ | $c_\phi$ | $c_{\phi D}$ | $c_{\phi\square}$ | $c^{(1)}_{\phi\psi_L}$ | $c^{(3)}_{\phi\psi_L}$ | $c_{\phi\psi_R}$ | $c_{\phi ud}$ | $c_{\psi_R\phi}$ |
|---|---|---|---|---|---|---|---|---|
| $c_\phi$ | 0 | 0 | 0 | 0 | 0 | 0 | 0 | 0 |
| $c_{\phi D}$ | | × | ∅ | 0 | 0 | 0 | 0 | 0 |
| $c_{\phi\square}$ | | | 0 | 0 | 0 | 0 | 0 | 0 |
| $c^{(1)}_{\phi\psi_L}$ | | | | × | 0 | 0 | 0 | 0 |
| $c^{(3)}_{\phi\psi_L}$ | | | | | × | 0 | 0 | 0 |
| $c_{\phi\psi_R}$ | | | | | | × | 0 | 0 |
| $c_{\phi ud}$ | | | | | | | × | 0 |
| $c_{\psi_R\phi}$ | | | | | | | | 0 |

| $\gamma'_{c^{(1)}_{B\phi^4 D^2}}$ | $c_\phi$ | $c_{\phi D}$ | $c_{\phi\square}$ | $c^{(1)}_{\phi\psi_L}$ | $c^{(3)}_{\phi\psi_L}$ | $c_{\phi\psi_R}$ | $c_{\phi ud}$ | $c_{\psi_R\phi}$ |
|---|---|---|---|---|---|---|---|---|
| $c_\phi$ | 0 | 0 | 0 | 0 | 0 | 0 | 0 | 0 |
| $c_{\phi D}$ | | × | 0 | 0 | 0 | 0 | 0 | 0 |
| $c_{\phi\square}$ | | | 0 | 0 | 0 | 0 | 0 | 0 |
| $c^{(1)}_{\phi\psi_L}$ | | | | × | 0 | 0 | 0 | 0 |
| $c^{(3)}_{\phi\psi_L}$ | | | | | × | 0 | 0 | 0 |
| $c_{\phi\psi_R}$ | | | | | | × | 0 | 0 |
| $c_{\phi ud}$ | | | | | | | × | 0 |
| $c_{\psi_R\phi}$ | | | | | | | | 0 |

All other $\gamma'$ matrices vanish identically, with all their zeros being trivial.

Finally, in Table 4 we provide a different view on the global structure of the anomalous dimensions, by showing, for each pair of dimension-six interactions, the dimension-eight operators that get renormalised by them. Despite being not explicitly shown, contributions proportional to two fermionic operators involve only leptons or quarks, but not both.

| | $c_\phi$ | $c_{\phi D}$ | $c_{\phi\Box}$ | $c^{(1)}_{\phi\psi_L}$ | $c^{(3)}_{\phi\psi_L}$ | $c_{\phi\psi_R}$ | $c_{\phi ud}$ | $c_{\psi_R\phi}$ |
|---|---|---|---|---|---|---|---|---|
| $c_\phi$ | $\phi^8$ | $\phi^8, \phi^{6(1)}, \phi^{6(2)}$ | $\phi^8, \phi^{6(1)}$ | | $\phi^8$ | | $\phi^8$ | $\phi^8$ |
| $c_{\phi D}$ | | $\phi^8, \phi^{6(1)}, \phi^{6(2)}$<br>$\phi^{4(1)}, \phi^{4(2)}, \phi^{4(3)}$<br>$V^2\phi^{4(1)}, V\phi^4 D^{2(1)}$ | $\phi^8, \phi^{6(1)}, \phi^{6(2)}$<br>$\phi^{4(1)}, \phi^{4(2)}, \phi^{4(3)}$ | $\phi^8, \phi^{6(1)}, \phi^{6(2)}$ | | $\phi^8, \phi^{6(1)}, \phi^{6(2)}$ | $\phi^8, \phi^{6(1)}, \phi^{6(2)}$ | $\phi^8, \phi^{6(1)}, \phi^{6(2)}$ |
| $c_{\phi\Box}$ | | | $\phi^8, \phi^{6(1)}, \phi^{6(2)}$<br>$\phi^{4(1)}, \phi^{4(2)}, \phi^{4(3)}$ | | $\phi^8, \phi^{6(1)}$ | | $\phi^8, \phi^{6(1)}$ | $\phi^8, \phi^{6(1)}$ |
| $c^{(1)}_{\phi\psi_L}$ | | | | $\phi^8, \phi^{6(1)}, \phi^{6(2)}$<br>$\phi^{4(1)}, \phi^{4(2)}$<br>$V^2\phi^{4(1)}, V\phi^4 D^{2(1)}$ | $\phi^8, \phi^{6(1)}, \phi^{6(2)}$ | $\phi^8, \phi^{6(1)}, \phi^{6(2)}$ | | $\phi^8, \phi^{6(1)}, \phi^{6(2)}$ |
| $c^{(3)}_{\phi\psi_L}$ | | | | | $\phi^8, \phi^{6(1)}, \phi^{6(2)}$<br>$\phi^{4(1)}, \phi^{4(2)}, \phi^{4(3)}$<br>$V^2\phi^{4(1)}, V\phi^4 D^{2(1)}$ | $\phi^8, \phi^{6(1)}, \phi^{6(2)}$ | $\phi^{6(1)}, \phi^{6(2)}$ | $\phi^8, \phi^{6(1)}$ |
| $c_{\phi\psi_R}$ | | | | | | $\phi^8, \phi^{6(1)}, \phi^{6(2)}$<br>$\phi^{4(1)}, \phi^{4(2)}$<br>$V^2\phi^{4(1)}, V\phi^4 D^{2(1)}$ | | $\phi^8, \phi^{6(1)}, \phi^{6(2)}$ |
| $c_{\phi ud}$ | | | | | | | $\phi^8, \phi^{6(1)}, \phi^{6(2)}$<br>$\phi^{4(1)}, \phi^{4(3)}$<br>$V^2\phi^{4(1)}, V\phi^4 D^{2(1)}$ | $\phi^{6(1)}, \phi^{6(2)}$ |
| $c_{\psi_R\phi}$ | | | | | | | | $\phi^8, \phi^{6(1)}, \phi^{6(2)}$ |

Table 4: *Dimension-eight operators that are renormalised by the pairs of dimension-six interactions in the corresponding column and row. $V$ stands for either $W$ or $B$.*

## 5   Discussion and outlook

We conclude this article highlighting several observations that can be made on the basis of the RGE matrices above:

**1.** All the dimension-eight operators that are renormalised can arise at tree level in UV completions of the SM [29]. The reason is simply that those operators that arise only at loop level involve two Higgs fields (see Table 1), unlike any one-loop diagram containing two insertions of the dimension-six terms. The same holds for dimension-six operators. Thus, we conclude that within the bosonic sector of the SMEFT, dimension-six tree-level operators do not mix into loop-level operators to order $E^4/\Lambda^4$. This extends previous findings at order $E^2/\Lambda^2$ [41].

**2.** Several of the $\gamma'$ matrices above exhibit a number of zeros (denoted by 0) for which all contributions in the Green basis are vanishing (they can result simply from the absence of Feynman diagrams or from CP conservation reasons). For example, the first row in $\gamma'_{c^{(1)}_{\phi^4}}$ reflects that there are no one-particle-irreducible diagrams with four Higgses involving the insertion of one six-Higgs operator and one four-Higgs interaction. Instead, those denoted by $\emptyset$ ensue from non-trivial cancellations between different counterterms in the Green basis which, on-shell, add to zero. For example, the (23) entry of $\gamma'_{c^{(1)}_{W^2\phi^4}}$ vanishes because the terms proportional to $c_{\phi D}c_{\phi\Box}$ in $\tilde{c}^{(1)}_{W^2\phi^4}$ and $\tilde{c}^{(7)}_{W\phi^4 D^2}$ cancel in Eq. (73). Zeros as this one might be understood on the basis of the helicity-amplitude formalism [54, 55].

**3.** Related to the previous point, we find the very surprising result that the Peskin-Takeuchi parameters [56] $S$ and $U$ are not renormalised by tree-level dimension-six operators to order $v^4/\Lambda^4$. Indeed, these observables read [17, 27]:

$$\frac{1}{16\pi}S = \frac{v^2}{\Lambda^2}\left[c_{\phi WB} + c^{(1)}_{WB\phi^4}\frac{v^4}{\Lambda^4}\right], \qquad \frac{1}{16\pi}U = \frac{v^4}{\Lambda^4}c^{(3)}_{W^2\phi^4}, \tag{40}$$

with $\mathcal{O}_{\phi WB} = (\phi^\dagger\sigma^I\phi)W^I_{\mu\nu}B^{\mu\nu}$. (Note that $U$ arises only at dimension eight.) What we find is that $c^{(1)}_{WB\phi^4}$ and $c^{(3)}_{W^2\phi^4}$ do not renormalise because the direct contribution cancels that from the redundant operator $\mathcal{O}^{(7)}_{W\phi^4 D^2}$. This fact, together with the non-renormalisation of $c_{\phi WB}$ found in Refs. [9], shows that both $S$ and $U$ are not triggered by dimension-six tree-level interactions at one loop.

**4.** The Wilson coefficients $c^{(1)}_{\phi^4}$, $c^{(2)}_{\phi^4}$ and $c^{(3)}_{\phi^4}$ are subject to positivity constraints [33]. In particular, $c^{(2)}_{\phi^4} \geq 0$, $c^{(1)}_{\phi^4} + c^{(2)}_{\phi^4} \geq 0$ and $c^{(1)}_{\phi^4} + c^{(2)}_{\phi^4} + c^{(3)}_{\phi^4} \geq 0$. These inequalities should reflect in the corresponding matrices.

To see how, let us first note that there exist well-behaved UV completions of the SM that induce, at tree level, the operators $c_\phi$, $c_{\phi D}$ and $c_{\phi\Box}$ with *arbitrary values*; see Appendix C for a particular example. The values of the dimension-eight Wilson coefficients $c^{(1)}_{\phi^4}$, $c^{(2)}_{\phi^4}$ and $c^{(3)}_{\phi^4}$ at any energy $\mu < M$ triggered by double insertions of the dimension-six operators scale differently with the model couplings than the tree-level contribution (which in general can not be avoided). In particular, within the model of Appendix C, we have $\sim \kappa^4/M^4$ versus $\sim \kappa^2/M^2$. This suggests that both contributions must satisfy the positivity bounds separately. Indeed, in the limit of scale-invariant dimension-six Wilson coefficients, we can check that:

$$16\pi^2 c^{(2)}_{\phi^4} = \frac{1}{3}(5c^2_{\phi D} + 16c_{\phi D}c_{\phi\Box} + 16c^2_{\phi\Box})\log\frac{M}{\mu} > 0, \tag{41}$$

$$16\pi^2 \left[ c_{\phi^4}^{(1)} + c_{\phi^4}^{(2)} \right] = \frac{16}{3}(c_{\phi D}^2 - c_{\phi D}c_{\phi\Box} + 2c_{\phi\Box}^2) \log \frac{M}{\mu} > 0 \,, \tag{42}$$

$$16\pi^2 \left[ c_{\phi^4}^{(1)} + c_{\phi^4}^{(2)} + c_{\phi^4}^{(3)} \right] = 3(c_{\phi D}^2 + 8c_{\phi\Box}^2) \log \frac{M}{\mu} > 0 \,; \tag{43}$$

for arbitrary values of $c_{\phi D}$ and $c_{\phi\Box}$. (Fermionic Wilson coefficients do not modify these relations because they contribute as sums of modulus squared and therefore positively, as a result also of very fine cancellations between positive and negative terms in Eqs. 17–19.) Note that these inequalities hold non-trivially; for example $c_{\phi^4}^{(1)}$ is negative in a neighbourhood of its minimum. It should be possible to extend this kind of analysis to other operators (which do not renormalise within the assumptions we make in this work), thus providing interesting cross-checks of the anomalous dimensions (or new bounds on Wilson coefficients).

**5.** Among the non-vanishing entries in the different $\gamma'$s, we find values that depart significantly from the naive estimate of $\mathcal{O}(1)$. The most notable of these, not suppressed by gauge or $\lambda$ couplings, are the 126 in $\gamma'_{\phi^8}$ and the 96 in $\gamma'_{c_{\phi^6}^{(1)}}$; see Appendix B. As we discuss below, these large numbers can have important low-energy implications.

**6.** Although we do not aim to exhaust all possible phenomenological implications of the running of the dimension-eight operators, we would like to stress that the $T$ parameter, defined by [17]

$$\alpha T = -\frac{1}{2} \frac{v^2}{\Lambda^2} \left[ c_{\phi D} + c_{\phi^6}^{(2)} \frac{v^2}{\Lambda^2} \right] \,, \tag{44}$$

with $\alpha \sim 1/137$ being the fine-structure constant, receives contributions from the operator $\mathcal{O}_{\phi ud}$ only at order $v^4/\Lambda^4$ (because $c_{\phi D}$ is not renormalised by one insertion of $\mathcal{O}_{\phi ud}$; see Ref. [8]). Using bounds on $T$ from Ref. [57], and assuming that only $c_{\phi tb}$ is non-vanishing, we obtain $c_{\phi tb} \leq 5.9$ for $\Lambda = 1$ TeV. This constraint is competitive with the value $c_{\phi tb} \leq 5.3$ reported in Ref. [58].

**7.** We would also like to emphasize the importance of one-loop $v^4/\Lambda^4$ effects for the EW phase transition (EWPT) ensuing from modifications of the Higgs potential [16, 59–63]. To this aim, let us assume that $c_\phi$ is the only non-vanishing Wilson coefficient in the UV, and let us neglect gauge and Yukawa couplings. The Higgs potential in the infrared is then provided by running $\mathcal{L}_{\text{UV}}$ down to the EW scale. In the leading-logarithm approximation, this reads:

$$V \sim \mu^2|\phi|^2 + \lambda|\phi|^4 + \frac{c_\phi}{\Lambda^2}\left(1 - \frac{108}{16\pi^2}\lambda \log\frac{\Lambda}{v}\right)|\phi|^6 + \frac{126}{16\pi^2\Lambda^4}\log\frac{\Lambda}{v}c_\phi^2\,|\phi|^8 \,, \tag{45}$$

where $c_\phi$ is evaluated in the UV, and the renormalisable couplings are assumed scale-invariant for simplicity. (The first logarithm can be read from Ref. [7].) In both $|\phi|^6$ and $|\phi|^8$ we have included only the dominant corrections.

Following Refs. [16, 64], we know that the EWPT is first order and strong as required by EW baryogenesis [65] provided that $500\,\text{GeV} \lesssim \Lambda/\sqrt{c_{\text{eff}}} \lesssim 750\,\text{GeV}$, where we have defined $c_{\text{eff}} = c_\phi + 3/2\,v^2/\Lambda^2 c_{\phi^8}$. Fixing, as a matter of example, $\Lambda = 1$ TeV, it can be easily checked that this occurs for:

$$1.7\,\text{TeV}^{-2} \lesssim c_\phi \lesssim 3.7\,\text{TeV}^{-2} \tag{46}$$

if the running of $c_{\phi^8}$ is neglected, whereas if we account for it we obtain:

$$1.5\,\text{TeV}^{-2} \lesssim c_\phi \lesssim 2.6\,\text{TeV}^{-2} \,. \tag{47}$$

| | $d_5$ | $d_5^2$ | $d_6$ | $d_5^3$ | $d_5 \times d_6$ | $d_7$ | $d_5^4$ | $d_5^2 \times d_6$ | $d_6^2$ | $d_5 \times d_7$ | $d_8$ |
|---|---|---|---|---|---|---|---|---|---|---|---|
| $d_{\leq 4}$ (bosonic) | | | ✓ [7] | | | | | | This work | | ✗ |
| $d_{\leq 4}$ (fermionic) | | | ✓ [7] | | | | | | ✗ | | ✗ |
| $d_5$ | ✓ [66–68] | | | | ✓ [71] | ✓ [71] | | | | | |
| $d_6$ (bosonic) | | ✓ [30] | ✓ [7–9] | | | | | ✗ | This work | ✗ | ✗ |
| $d_6$ (fermionic) | | ✓ [30] | ✓ [7–9, 69] | | | | | ✗ | ✗ | ✗ | ✗ |
| $d_7$ | | | | ✓ [71] | ✓ [71] | ✓ [22, 70] | | | | | |
| $d_8$ (bosonic) | | | | | | | ✗ | ✗ | This work | ✗ | ✗ |
| $d_8$ (fermionic) | | | | | | | ✗ | ✗ | ✗ | ✗ | ✗ |

Table 5: *State of the art of SMEFT renormalisation. The rows represent the operators (defined by their dimension d) being renormalised, while the columns show the operators entering the loops. Note that there are no bosonic interactions at odd dimension. Blank entries vanish. A tick ✓ represents that the complete contribution is known. The ✓ indicates that only (but substantial) partial results have been already obtained. The ✗ indicates that nothing, or very little, is known. The contribution made in this paper is marked by ▬.*

The 30 % difference in the upper limit evidences the potential importance of both dimension-eight operators and their running.

To conclude, let us remark that our results comprise one step further towards the one-loop renormalisation of the SMEFT to order $v^4/\Lambda^4$. This endeavor was initiated in Refs. [4–9] (see also Refs. [66–68] for the renormalisation of the Weinbeg operator) and continued in Ref. [69] (for baryon-number violating interactions), Ref. [30] (which includes the renormalisation of dimension-six operators by pairs of Weinberg interactions), Refs. [22, 70] (which involves the renormalisation of dimension-seven operators triggered by relevant couplings) and Ref. [71] (in which neutrino masses are renormalised to order $v^3/\Lambda^3$, including arbitrary combinations of dimension-five, -six and -seven operators). Some partial results of renormalisation within the dimension-eight sector of the SMEFT can be found in Refs. [19, 72]. See Table 5 for a summary of the state of the art.

In forecoming works, we plan to extend the results of this paper with the inclusion, in the UV, of the operators of dimension eight that can be generated at tree level. We will also consider the renormalisation of non-bosonic operators. The latter can be induced, in particular, by field redefinitions aimed at removing the operators $\mathcal{O}_{BD\phi}$, $\mathcal{O}'_{\phi D}$, $\mathcal{O}_{WD\phi}$ and $\mathcal{O}^{(6)}_{W\phi^4 D^2}$; see Appendix A. Consequently, our current findings lay the basis for future work in this direction.

# Acknowledgments

We would like to thank Jorge de Blas for useful discussions. MC and JS are supported by the Ministry of Science and Innovation under grant number FPA2016-78220-C3-1/3-P (FEDER), SRA under grant PID2019-106087GB-C21/C22 (10.13039/501100011033), and by the Junta de Andalucía grants FQM 101, A-FQM-211-UGR18 and P18-FR-4314 (FEDER). MC is also supported by the Spanish MINECO under the Ramón y Cajal programme. GG and MR acknowledge support by LIP (FCT, COMPETE2020-Portugal2020, FEDER, POCI-01-0145-FEDER-007334) as well as by FCT under project CERN/FIS-PAR/0024/2019. GG is also supported by FCT under the grant SFRH/BD/144244/2019. MR is also supported by Fundação para a Ciência e Tecnologia (FCT) under the grant PD/BD/142773/2018.

## A  Removing redundant operators

The redundant operators generated in the process of renormalisation can be removed upon performing suitable perturbative field redefinitions, for example $\phi \to \phi + \frac{1}{\Lambda^2}\mathcal{O}$, where $\mathcal{O}$ is called the *perturbation*. We are interested in the effect of these field redefinitions to linear order in the perturbation (because $\mathcal{O}$ is loop suppressed and therefore quadratic powers of this term are formally two-loop corrections), which can be implemeted through the equations of motion of the SMEFT to order $v^2/\Lambda^2$ [53]. These read [73]:

$$
D^2\phi^i = -\mu^2\phi^i - 2\lambda(\phi^\dagger\phi)\phi^i + \frac{1}{\Lambda^2}\Bigg\{ 3c_\phi(\phi^\dagger\phi)^2\phi^i + 2c_{\phi\square}\phi^i\square(\phi^\dagger\phi)
$$
$$
- c_{\phi D}\left[ (D^\mu\phi)^i\left(\phi^\dagger\overleftrightarrow{D}_\mu\phi\right) + \phi^i\partial^\mu\left(\phi^\dagger D_\mu\phi\right)\right]\Bigg\} + \cdots , \tag{48}
$$

$$
\partial^\nu B_{\mu\nu} = \frac{g_1}{2}\phi^\dagger i\overleftrightarrow{D}_\mu\phi + \frac{c_{\phi D}}{\Lambda^2}\frac{g_1}{2}\left(\phi^\dagger\phi\right)\left(\phi^\dagger i\overleftrightarrow{D}_\mu\phi\right) + \cdots , \tag{49}
$$

$$
D^\nu W^I_{\mu\nu} = \frac{g_2}{2}\phi^\dagger i\overleftrightarrow{D}^I_\mu\phi + \frac{c_{\phi D}}{\Lambda^2}g_2\left(\phi^\dagger\sigma^I\phi\right)\left(\phi^\dagger i\overleftrightarrow{D}_\mu\phi\right) + \cdots , \tag{50}
$$

where the ellipses represent fermionc operators, on which we are not interested. The following relations hold on-shell:

$$
\mathcal{O}_{BD\phi} = \frac{g_1}{2}\left[\mathcal{O}_{\phi\square} + 4\mathcal{O}_{\phi D} - \frac{\mu^2}{\Lambda^2}c_{\phi D}\mathcal{O}_\phi\right] + \frac{g_1}{2}\frac{c_{\phi D}}{\Lambda^2}\left[ -2\lambda\mathcal{O}_{\phi 8} + 3\mathcal{O}^{(1)}_{\phi 6} + 2\mathcal{O}^{(2)}_{\phi 6}\right] + \cdots , \tag{51}
$$

$$
\mathcal{O}'_{\phi D} = -\frac{1}{2}\left[ -\mathcal{O}_{\phi\square} - 2\mu^2|\phi|^4 - 4\lambda\mathcal{O}_\phi + \frac{\mu^2}{\Lambda^2}\left(c_{\phi D} - 8c_{\phi\square}\right)\mathcal{O}_\phi\right]
$$
$$
- \frac{1}{2\Lambda^2}\left[ \left(6c_\phi - 16\lambda c_{\phi\square} + 2\lambda c_{\phi D}\right)\mathcal{O}_{\phi 8} + \left(8c_{\phi\square} + c_{\phi D}\right)\mathcal{O}^{(1)}_{\phi 6} + 2c_{\phi D}\mathcal{O}^{(2)}_{\phi 6}\right] + \cdots , \tag{52}
$$

$$
\mathcal{O}_{WD\phi} = -g_2\left[ -\frac{3}{2}\mathcal{O}_{\phi\square} - 2\mu^2|\phi|^4 - 4\lambda\mathcal{O}_\phi + \frac{\mu^2}{\Lambda^2}\left(2c_{\phi D} - 8c_{\phi\square}\right)\mathcal{O}_\phi\right]
$$
$$
- \frac{g_2}{\Lambda^2}\left[ \left(6c_\phi - 16\lambda c_{\phi\square} + 4\lambda c_{\phi D}\right)\mathcal{O}_{\phi 8} + \left(8c_{\phi\square} - 2c_{\phi D}\right)\mathcal{O}^{(1)}_{\phi 6}\right] + \cdots ; \tag{53}
$$

where the ellipses encode again fermionic interactions. The operator $\mathcal{O}''_{\phi D}$ gives no contributions to the bosonic sector.

To arrive at these results, we used the following identities:

$$
|\phi|^2\left(\phi^\dagger D_\mu\phi\right)\left(D^\mu\phi^\dagger\phi\right) = \frac{1}{2}\left[\mathcal{O}^{(1)}_{\phi 6} + \mathcal{O}^{(2)}_{\phi 6}\right] , \tag{54}
$$

$$
|\phi|^2\left(\phi^\dagger i\overleftrightarrow{D}\phi\right)^2 = 3\mathcal{O}^{(1)}_{\phi 6} + 2\mathcal{O}^{(2)}_{\phi 6} + \frac{1}{2}\mathcal{O}^{(3)}_{\phi 6} , \tag{55}
$$

$$
|\phi|^4\square|\phi|^2 = 2\mathcal{O}^{(1)}_{\phi 6} + \mathcal{O}^{(3)}_{\phi 6} . \tag{56}
$$

In turn, the redundant dimension-eight operators become, on-shell:

$$
\mathcal{O}^{(3)}_{\phi 6} = -2\mu^2\mathcal{O}_\phi - 4\lambda\mathcal{O}_{\phi 8} , \tag{57}
$$

$$
\mathcal{O}^{(4)}_{\phi 4} = \mu^2\left[ -\mathcal{O}_{\phi D} - 2\mu^2|\phi|^4 - 4\lambda\mathcal{O}_\phi\right] - 4\lambda\mathcal{O}^{(1)}_{\phi 6} , \tag{58}
$$

$$\mathcal{O}_{\phi^4}^{(6)} = -2\mu^2 \mathcal{O}_{\phi D} - 2\lambda \left[ \mathcal{O}_{\phi^6}^{(1)} + \mathcal{O}_{\phi^6}^{(2)} \right] , \tag{59}$$

$$\mathcal{O}_{\phi^4}^{(8)} = 2\mu^2 \left[ \mu^2 |\phi|^4 + 4\lambda \mathcal{O}_\phi \right] + 8\lambda^2 \mathcal{O}_{\phi^8} , \tag{60}$$

$$\mathcal{O}_{\phi^4}^{(10)} = \mu^2 \left[ \mu^2 |\phi|^4 + 4\lambda \mathcal{O}_\phi \right] + 4\lambda^2 \mathcal{O}_{\phi^8} , \tag{61}$$

$$\mathcal{O}_{\phi^4}^{(11)} = \mu^2 \left[ \mu^2 |\phi|^4 + 4\lambda \mathcal{O}_\phi \right] + 4\lambda^2 \mathcal{O}_{\phi^8} , \tag{62}$$

$$\mathcal{O}_{\phi^4}^{(12)} = \mu^2 \left[ 2\mathcal{O}_{\phi D} + \mathcal{O}_{\phi \Box} - 2\lambda \mathcal{O}_\phi \right] + 2 \left[ 2\lambda \mathcal{O}_{\phi^6}^{(1)} + \lambda \mathcal{O}_{\phi^6}^{(2)} - 2\lambda^2 \mathcal{O}_{\phi^8} \right] , \tag{63}$$

$$\mathcal{O}_{W\phi^4 D^2}^{(6)} = -\frac{g_2}{2} \left[ -\mu^2 \mathcal{O}_\phi + 5\mathcal{O}_{\phi^6}^{(1)} - 2\lambda \mathcal{O}_{\phi^8} + \cdots \right] , \tag{64}$$

$$\mathcal{O}_{W\phi^4 D^2}^{(7)} = \frac{1}{4} \left\{ \frac{g_2}{2} \left[ 2\mu^2 \mathcal{O}_\phi + \mathcal{O}_{W^2 \phi^4}^{(1)} + \mathcal{O}_{W^2 \phi^4}^{(3)} \right] + g_1 \mathcal{O}_{WB\phi^4}^{(1)} + 8\mathcal{O}_{W\phi^4 D^2}^{(1)} \right.$$
$$\left. - 6g_2 \mathcal{O}_{\phi^6}^{(1)} + g_2 \mathcal{O}_{\phi^6}^{(2)} + 2g_2 \lambda \mathcal{O}_{\phi^8}^{(1)} \right\} , \tag{65}$$

$$\mathcal{O}_{B\phi^4 D^2}^{(3)} = -\frac{g_1}{2} \left[ -\mu^2 \mathcal{O}_\phi + 3\mathcal{O}_{\phi^6}^{(1)} + 2\mathcal{O}_{\phi^6}^{(2)} - 2\lambda \mathcal{O}_{\phi^8} \right] ; \tag{66}$$

where again the ellipses represent terms on which we are not interested in this work. The operator $\mathcal{O}_{\phi^6}^{(4)}$ contributes only to fermionic interactions.

Altogether, these equations lead to:

$$\lambda \rightarrow \lambda - \frac{\mu^2}{\Lambda^2} c_{\phi D}' - \frac{\mu^4}{\Lambda^4} \left[ -2 \left( c_{\phi^4}^{(4)} - c_{\phi^4}^{(8)} \right) + c_{\phi^4}^{(10)} + c_{\phi^4}^{(11)} \right] , \tag{67}$$

$$c_\phi \rightarrow c_\phi + 2\lambda c_{\phi D}' + \frac{\mu^2}{2\Lambda^4} \left[ 3(c_{\phi D} + 2c_{\phi \Box}) c_\phi - (c_{\phi D} - 8c_{\phi \Box})(c_{\phi D}' + 2g_2 c_{WD\phi}) \right.$$
$$- 2g_2 c_{\phi D} c_{WD\phi} - g_1 c_{\phi D} c_{BD\phi} - 4c_{\phi^6}^{(3)} + 4\lambda \left\{ -2c_{\phi^4}^{(4)} + 4c_{\phi^4}^{(8)} + 2c_{\phi^4}^{(10)} + 2c_{\phi^4}^{(11)} - c_{\phi^4}^{(12)} \right\}$$
$$\left. + g_2 c_{W\phi^4 D^2}^{(6)} + \frac{g_2}{2} c_{W\phi^4 D^2}^{(7)} + g_1 c_{B\phi^4 D^2}^{(3)} \right] , \tag{68}$$

$$c_{\phi D} \rightarrow c_{\phi D} + 2\frac{\mu^2}{\Lambda^4} \left[ \frac{1}{2}(c_{\phi D} + 2c_{\phi \Box}) c_{\phi D} - c_{\phi^4}^{(6)} + c_{\phi^4}^{(12)} \right] , \tag{69}$$

$$c_{\phi \Box} \rightarrow c_{\phi \Box} + \frac{1}{2} c_{\phi D}' + \frac{\mu^2}{\Lambda^4} \left[ (c_{\phi D} + 2c_{\phi \Box}) c_{\phi \Box} - c_{\phi^4}^{(4)} + c_{\phi^4}^{(12)} \right] , \tag{70}$$

where we have already normalised canonically the Higgs kinetic term; as well as

$$c_{\phi^8} \rightarrow c_{\phi^8} - g_1 \lambda c_{BD\phi} c_{\phi D} - \left( c_{\phi D}' + 2g_2 c_{WD\phi} \right) \left( 3c_\phi - 8\lambda c_{\phi \Box} + \lambda c_{\phi D} \right) \tag{71}$$
$$- 2g_2 \lambda c_{WD\phi} c_{\phi D} - 4\lambda c_{\phi^6}^{(3)} + 4\lambda^2 \left[ 2c_{\phi^4}^{(8)} + c_{\phi^4}^{(10)} + c_{\phi^4}^{(11)} - c_{\phi^4}^{(12)} \right]$$
$$+ g_2 \lambda c_{W\phi^4 D^2}^{(6)} + \frac{g_2}{2} \lambda c_{W\phi^4 D^2}^{(7)} + g_1 \lambda c_{B\phi^4 D^2}^{(3)} ,$$

$$c_{\phi^6}^{(1)} \rightarrow c_{\phi^6}^{(1)} + \frac{3}{2} g_1 c_{BD\phi} c_{\phi D} - \frac{\left( c_{\phi D}' + 2g_2 c_{WD\phi} \right)}{2} (8c_{\phi \Box} + c_{\phi D}) + 3g_2 c_{WD\phi} c_{\phi D} \tag{72}$$
$$- 2\lambda \left[ 2c_{\phi^4}^{(4)} + c_{\phi^4}^{(6)} - 2c_{\phi^4}^{(12)} \right] - \frac{5}{2} g_2 c_{W\phi^4 D^2}^{(6)} - \frac{3}{2} g_2 c_{W\phi^4 D^2}^{(7)} - \frac{3}{2} g_1 c_{B\phi^4 D^2}^{(3)} ,$$

$$c_{\phi^6}^{(2)} \rightarrow c_{\phi^6}^{(2)} + g_1 c_{BD\phi} c_{\phi D} - c_{\phi D}' c_{\phi D} - 2\lambda \left[ c_{\phi^4}^{(6)} - c_{\phi^4}^{(12)} \right] + \frac{g_2}{4} c_{W\phi^4 D^2}^{(7)} - g_1 c_{B\phi^4 D^2}^{(3)} ,$$

$$c_{W^2\phi^4}^{(1)} \to c_{W^2\phi^4}^{(1)} + \frac{g_2}{8} c_{W\phi^4 D^2}^{(7)} , \tag{73}$$

$$c_{W^2\phi^4}^{(3)} \to c_{W^2\phi^4}^{(3)} + \frac{g_2}{8} c_{W\phi^4 D^2}^{(7)} , \tag{74}$$

$$c_{WB\phi^4}^{(1)} \to c_{WB\phi^4}^{(1)} + \frac{g_1}{4} c_{W\phi^4 D^2}^{(7)} \tag{75}$$

$$c_{W\phi^4 D^2}^{(1)} \to c_{W\phi^4 D^2}^{(1)} + 2 c_{W\phi^4 D^2}^{(7)} . \tag{76}$$

# B  Renormalisation group equations

For a given coupling $c$, we define:

$$\dot{c} \equiv 16\pi^2 \mu \frac{dc}{d\mu}. \tag{77}$$

Thus, we have:

$$\dot{\lambda} \supset \left( 5 c_{\phi D}^2 - 24 c_{\phi D} c_{\phi\Box} + 24 c_{\phi\Box}^2 \right) \frac{\mu^4}{\Lambda^4} , \tag{78}$$

$$
\begin{aligned}
\dot{c}_\phi \supset \bigg\{ &\frac{-37}{24} g_1^2 c_{\phi D}^2 + \frac{5}{8} g_2^2 c_{\phi D}^2 - 35\lambda c_{\phi D}^2 + \frac{26}{3} c_{\phi\Box} g_1^2 c_{\phi D} - \frac{113}{6} c_{\phi\Box} g_2^2 c_{\phi D} - 51 c_\phi c_{\phi D} \\
&+ 320\lambda c_{\phi\Box} c_{\phi D} - 3\text{Tr}[c_{d\phi} y^{d\dagger}] c_{\phi D} - \text{Tr}[c_{e\phi} y^{e\dagger}] c_{\phi D} - 3\text{Tr}[c_{u\phi} y^{u\dagger}] c_{\phi D} - 3\text{Tr}[(c_{d\phi})^\dagger y^d] c_{\phi D} \\
&- \text{Tr}[(c_{e\phi})^\dagger y^e] c_{\phi D} - 3\text{Tr}[(c_{u\phi})^\dagger y^u] c_{\phi D} - 6\text{Tr}[y^{u\dagger} y^d (c_{\phi ud})^\dagger] c_{\phi D} + 4\text{Tr}[c_{\phi l}^{(3)} y^e y^{e\dagger}] c_{\phi D} \\
&+ 12\text{Tr}[c_{\phi q}^{(3)} y^d y^{d\dagger}] c_{\phi D} + 12\text{Tr}[c_{\phi q}^{(3)} y^u y^{u\dagger}] c_{\phi D} - 6\text{Tr}[c_{\phi ud} y^{d\dagger} y^u] c_{\phi D} + \frac{2}{3} g_1^2 \text{Tr}[c_{\phi d}] c_{\phi D} \\
&+ \frac{2}{3} g_1^2 \text{Tr}[c_{\phi e}] c_{\phi D} + \frac{2}{3} g_1^2 \text{Tr}[c_{\phi l}^{(1)}] c_{\phi D} - \frac{8}{3} g_2^2 \text{Tr}[c_{\phi l}^{(3)}] c_{\phi D} - \frac{2}{3} g_1^2 \text{Tr}[c_{\phi q}^{(1)}] c_{\phi D} - 8 g_2^2 \text{Tr}[c_{\phi q}^{(3)}] c_{\phi D} \\
&- \frac{4}{3} g_1^2 \text{Tr}[c_{\phi u}] c_{\phi D} - 640\lambda c_{\phi\Box}^2 - \frac{20}{3} c_{\phi\Box}^2 g_1^2 + 20 c_{\phi\Box}^2 g_2^2 + 222 c_\phi c_{\phi\Box} + 24 c_{\phi\Box} \text{Tr}[c_{d\phi} y^{d\dagger}] \\
&+ 8 c_{\phi\Box} \text{Tr}[c_{e\phi} y^{e\dagger}] + 24 c_{\phi\Box} \text{Tr}[c_{u\phi} y^{u\dagger}] + 24 c_{\phi\Box} \text{Tr}[(c_{d\phi})^\dagger y^d] - 12\text{Tr}[(c_{d\phi})^\dagger c_{d\phi}] \\
&+ 8 c_{\phi\Box} \text{Tr}[(c_{e\phi})^\dagger y^e] - 4\text{Tr}[(c_{e\phi})^\dagger c_{e\phi}] + 24 c_{\phi\Box} \text{Tr}[(c_{u\phi})^\dagger y^u] - 12\text{Tr}[(c_{u\phi})^\dagger c_{u\phi}] \\
&- 2 g_2^2 \text{Tr}[(c_{\phi d})^\dagger c_{\phi d}] - \frac{2}{3} g_2^2 \text{Tr}[(c_{\phi e})^\dagger c_{\phi e}] - \frac{4}{3} g_2^2 \text{Tr}[(c_{\phi l}^{(1)})^\dagger c_{\phi l}^{(1)}] - \frac{8}{3} g_1^2 \text{Tr}[(c_{\phi l}^{(3)})^\dagger c_{\phi l}^{(3)}] \\
&+ \frac{4}{3} g_2^2 \text{Tr}[(c_{\phi l}^{(3)})^\dagger c_{\phi l}^{(3)}] - 4 g_2^2 \text{Tr}[(c_{\phi q}^{(1)})^\dagger c_{\phi q}^{(1)}] - 8 g_1^2 \text{Tr}[(c_{\phi q}^{(3)})^\dagger c_{\phi q}^{(3)}] + 4 g_2^2 \text{Tr}[(c_{\phi q}^{(3)})^\dagger c_{\phi q}^{(3)}] \\
&- 2 g_2^2 \text{Tr}[(c_{\phi u})^\dagger c_{\phi u}] + g_1^2 \text{Tr}[(c_{\phi ud})^\dagger c_{\phi ud}] - g_2^2 \text{Tr}[(c_{\phi ud})^\dagger c_{\phi ud}] + 48 c_{\phi\Box} \text{Tr}[y^{u\dagger} y^d (c_{\phi ud})^\dagger] \\
&- 6\text{Tr}[c_{d\phi} y^{d\dagger} c_{\phi q}^{(1)}] - 6\text{Tr}[c_{d\phi} y^{d\dagger} c_{\phi q}^{(3)}] - 2\text{Tr}[c_{e\phi} y^{e\dagger} c_{\phi l}^{(1)}] - 2\text{Tr}[c_{e\phi} y^{e\dagger} c_{\phi l}^{(3)}] + 6\text{Tr}[c_{u\phi} y^{u\dagger} c_{\phi q}^{(1)}] \\
&- 6\text{Tr}[c_{u\phi} y^{u\dagger} c_{\phi q}^{(3)}] + 6\text{Tr}[c_{\phi d} y^{d\dagger} c_{d\phi}] + 2\text{Tr}[c_{\phi e} y^{e\dagger} c_{e\phi}] - 2\text{Tr}[c_{\phi l}^{(1)} y^e (c_{e\phi})^\dagger] - 32 c_{\phi\Box} \text{Tr}[c_{\phi l}^{(3)} y^e y^{e\dagger}] \\
&- 2\text{Tr}[c_{\phi l}^{(3)} y^e (c_{e\phi})^\dagger] - 6\text{Tr}[c_{\phi q}^{(1)} y^d (c_{d\phi})^\dagger] + 6\text{Tr}[c_{\phi q}^{(1)} y^u (c_{u\phi})^\dagger] - 96 c_{\phi\Box} \text{Tr}[c_{\phi q}^{(3)} y^d y^{d\dagger}] \\
&- 6\text{Tr}[c_{\phi q}^{(3)} y^d (c_{d\phi})^\dagger] - 96 c_{\phi\Box} \text{Tr}[c_{\phi q}^{(3)} y^u y^{u\dagger}] - 6\text{Tr}[c_{\phi q}^{(3)} y^u (c_{u\phi})^\dagger] - 6\text{Tr}[c_{\phi u} y^{u\dagger} c_{u\phi}] \\
&+ 48 c_{\phi\Box} \text{Tr}[c_{\phi ud} y^{d\dagger} y^u] + 6\text{Tr}[(c_{d\phi})^\dagger y^d c_{\phi d}] + 2\text{Tr}[(c_{e\phi})^\dagger y^e c_{\phi e}] - 6\text{Tr}[(c_{u\phi})^\dagger y^u c_{\phi u}] \\
&+ 6\text{Tr}[c_{\phi d} y^{d\dagger} y^d c_{\phi d}] + 2\text{Tr}[c_{\phi e} y^{e\dagger} y^e c_{\phi e}] + 2\text{Tr}[c_{\phi l}^{(1)} y^e y^{e\dagger} c_{\phi l}^{(1)}] + 2\text{Tr}[c_{\phi l}^{(1)} y^e y^{e\dagger} c_{\phi l}^{(3)}] \\
&- 4\text{Tr}[c_{\phi l}^{(1)} y^e c_{\phi e} y^{e\dagger}] + 2\text{Tr}[c_{\phi l}^{(3)} y^e y^{e\dagger} c_{\phi l}^{(1)}] + 2\text{Tr}[c_{\phi l}^{(3)} y^e y^{e\dagger} c_{\phi l}^{(3)}] - 4\text{Tr}[c_{\phi l}^{(3)} y^e c_{\phi e} y^{e\dagger}] \\
&+ 6\text{Tr}[c_{\phi q}^{(1)} y^d y^{d\dagger} c_{\phi q}^{(1)}] + 6\text{Tr}[c_{\phi q}^{(1)} y^d y^{d\dagger} c_{\phi q}^{(3)}] - 12\text{Tr}[c_{\phi q}^{(1)} y^d c_{\phi d} y^{d\dagger}] + 6\text{Tr}[c_{\phi q}^{(1)} y^u y^{u\dagger} c_{\phi q}^{(1)}]
\end{aligned}
\tag{79}
$$

$$- 6\mathrm{Tr}[c_{\phi q}^{(1)} y^u y^{u\dagger} c_{\phi q}^{(3)}] - 12\mathrm{Tr}[c_{\phi q}^{(1)} y^u c_{\phi u} y^{u\dagger}] + 6\mathrm{Tr}[c_{\phi q}^{(3)} y^d y^{d\dagger} c_{\phi q}^{(1)}] + 6\mathrm{Tr}[c_{\phi q}^{(3)} y^d y^{d\dagger} c_{\phi q}^{(3)}]$$

$$- 12\mathrm{Tr}[c_{\phi q}^{(3)} y^d c_{\phi d} y^{d\dagger}] - 6\mathrm{Tr}[c_{\phi q}^{(3)} y^u y^{u\dagger} c_{\phi q}^{(1)}] + 6\mathrm{Tr}[c_{\phi q}^{(3)} y^u y^{u\dagger} c_{\phi q}^{(3)}] + 12\mathrm{Tr}[c_{\phi q}^{(3)} y^u c_{\phi u} y^{u\dagger}]$$

$$+ 6\mathrm{Tr}[c_{\phi u} y^{u\dagger} y^u c_{\phi u}] + \frac{32}{3} c_{\phi\Box} g_2^2 \mathrm{Tr}[c_{\phi l}^{(3)}] + 32 c_{\phi\Box} g_2^2 \mathrm{Tr}[c_{\phi q}^{(3)}] \Big\} \frac{\mu^2}{\Lambda^2} \,,$$

$$\dot{c}_{\phi D} \supset \left( -10 c_{\phi D}^2 + 4 c_{\phi D} c_{\phi\Box} \right) \frac{\mu^2}{\Lambda^2} \,, \tag{80}$$

$$\dot{c}_{\phi\Box} \supset \left( -\frac{3}{2} c_{\phi D}^2 - 14 c_{\phi D} c_{\phi\Box} + 36 c_{\phi\Box}^2 \right) \frac{\mu^2}{\Lambda^2} \,. \tag{81}$$

312

$$\dot{c}_{\phi^8} = -126 c_\phi^2 - 60 c_{\phi D}^2 \lambda^2 - 864 c_{\phi\Box}^2 \lambda^2 - 4(c_{\phi D}^2 - 10 c_{\phi\Box} c_{\phi D} + 32 c_{\phi\Box}^2) \lambda^2 \tag{82}$$

$$+ 48 c_\phi (15 c_{\phi D} + 18 \lambda c_{\phi\Box} - 4 c_{\phi\Box}) \lambda + \frac{1}{6} g_2^2 \Big( c_{\phi D}^2 - 12 c_{\phi\Box} c_{\phi D} - 24 \mathrm{Tr}[(c_{\phi d})^\dagger c_{\phi d}]$$

$$- 8 \mathrm{Tr}[(c_{\phi e})^\dagger c_{\phi e}] - 16 \mathrm{Tr}[(c_{\phi l}^{(1)})^\dagger c_{\phi l}^{(1)}] - 48 \mathrm{Tr}[(c_{\phi q}^{(1)})^\dagger c_{\phi q}^{(1)}] - 24 \mathrm{Tr}[(c_{\phi u})^\dagger c_{\phi u}]$$

$$+ 12 \mathrm{Tr}[(c_{\phi ud})^\dagger c_{\phi ud}] \Big) \lambda + \frac{1}{12} g_1^2 \Big( 3 c_{\phi D}^2 - 4 c_{\phi\Box} c_{\phi D} - 16 c_{\phi\Box}^2 - 64 \mathrm{Tr}[(c_{\phi l}^{(3)})^\dagger c_{\phi l}^{(3)}]$$

$$- 192 \mathrm{Tr}[(c_{\phi q}^{(3)})^\dagger c_{\phi q}^{(3)}] + 24 \mathrm{Tr}[(c_{\phi ud})^\dagger c_{\phi ud}] \Big) \lambda + \frac{1}{12} g_2^2 \Big( - 5 c_{\phi D}^2 + 20 c_{\phi\Box} c_{\phi D} - 16 c_{\phi\Box}^2$$

$$+ 32 \mathrm{Tr}[(c_{\phi l}^{(3)})^\dagger c_{\phi l}^{(3)}] + 96 \mathrm{Tr}[(c_{\phi q}^{(3)})^\dagger c_{\phi q}^{(3)}] - 48 \mathrm{Tr}[(c_{\phi ud})^\dagger c_{\phi ud}] \Big) \lambda + \frac{1}{2} \Big\{ - 3(g_2^2 + 4\lambda) c_{\phi D}^2$$

$$- 4 c_{\phi\Box} (3 g_1^2 + 3 g_2^2 - 112 \lambda) c_{\phi D} - 48 c_\phi (c_{\phi D} - 9 c_{\phi\Box}) - 8 c_{\phi\Box}^2 (128 \lambda + 3 g_1^2 + 3 g_2^2)$$

$$+ 8 \Big( - 6 \mathrm{Tr}[(c_{d\phi})^\dagger c_{d\phi}] - 2 \mathrm{Tr}[(c_{e\phi})^\dagger c_{e\phi}] - 6 \mathrm{Tr}[(c_{u\phi})^\dagger c_{u\phi}] - 3 \mathrm{Tr}[y^d (c_{d\phi})^\dagger c_{\phi q}^{(1)}]$$

$$- 3 \mathrm{Tr}[y^d (c_{d\phi})^\dagger c_{\phi q}^{(3)}] + 3 \mathrm{Tr}[y^{d\dagger} c_{d\phi} c_{\phi d}] - \mathrm{Tr}[y^e (c_{e\phi})^\dagger c_{\phi l}^{(1)}] - \mathrm{Tr}[y^e (c_{e\phi})^\dagger c_{\phi l}^{(3)}] + \mathrm{Tr}[y^{e\dagger} c_{e\phi} c_{\phi e}]$$

$$- 3 \mathrm{Tr}[y^u c_{\phi u} (c_{u\phi})^\dagger] + 3 \mathrm{Tr}[y^{u\dagger} c_{\phi q}^{(1)} c_{u\phi}] - 3 \mathrm{Tr}[y^{u\dagger} c_{\phi q}^{(3)} c_{u\phi}] - 3 \mathrm{Tr}[c_{u\phi} c_{\phi u} y^{u\dagger}] + 3 \mathrm{Tr}[c_{\phi d} (c_{d\phi})^\dagger y^d]$$

$$+ \mathrm{Tr}[c_{\phi e} (c_{e\phi})^\dagger y^e] - \mathrm{Tr}[c_{\phi l}^{(1)} c_{e\phi} y^{e\dagger}] - \mathrm{Tr}[c_{\phi l}^{(3)} c_{e\phi} y^{e\dagger}] - 3 \mathrm{Tr}[c_{\phi q}^{(1)} c_{d\phi} y^{d\dagger}] - 3 \mathrm{Tr}[c_{\phi q}^{(3)} c_{d\phi} y^{d\dagger}]$$

$$+ 3 \mathrm{Tr}[(c_{u\phi})^\dagger c_{\phi q}^{(1)} y^u] - 3 \mathrm{Tr}[(c_{u\phi})^\dagger c_{\phi q}^{(3)} y^u] - 6 \mathrm{Tr}[y^d c_{\phi d} y^{d\dagger} c_{\phi q}^{(1)}] - 6 \mathrm{Tr}[y^d c_{\phi d} y^{d\dagger} c_{\phi q}^{(3)}]$$

$$+ 3 \mathrm{Tr}[y^d c_{\phi d} c_{d\phi} y^{d\dagger}] - 2 \mathrm{Tr}[y^e c_{\phi e} y^{e\dagger} c_{\phi l}^{(1)}] - 2 \mathrm{Tr}[y^e c_{\phi e} y^{e\dagger} c_{\phi l}^{(3)}] + \mathrm{Tr}[y^e c_{\phi e} c_{e\phi} y^{e\dagger}]$$

$$+ \mathrm{Tr}[y^{e\dagger} c_{\phi l}^{(1)} c_{\phi l}^{(1)} y^e] + \mathrm{Tr}[y^{e\dagger} c_{\phi l}^{(1)} c_{\phi l}^{(3)} y^e] + \mathrm{Tr}[y^{e\dagger} c_{\phi l}^{(3)} c_{\phi l}^{(1)} y^e] + \mathrm{Tr}[y^{e\dagger} c_{\phi l}^{(3)} c_{\phi l}^{(3)} y^e]$$

$$+ 3 (\mathrm{Tr}[y^{d\dagger} c_{\phi q}^{(1)} c_{\phi q}^{(1)} y^d] + \mathrm{Tr}[y^{d\dagger} c_{\phi q}^{(1)} c_{\phi q}^{(3)} y^d] + \mathrm{Tr}[y^{d\dagger} c_{\phi q}^{(3)} c_{\phi q}^{(1)} y^d] + \mathrm{Tr}[y^{d\dagger} c_{\phi q}^{(3)} c_{\phi q}^{(3)} y^d]$$

$$+ \mathrm{Tr}[y^u c_{\phi u} c_{\phi u} y^{u\dagger}] - 2 \mathrm{Tr}[y^{u\dagger} c_{\phi q}^{(1)} y^u c_{\phi u}] + \mathrm{Tr}[y^{u\dagger} c_{\phi q}^{(1)} c_{\phi q}^{(1)} y^u] - \mathrm{Tr}[y^{u\dagger} c_{\phi q}^{(1)} c_{\phi q}^{(3)} y^u]$$

$$+ 2 \mathrm{Tr}[y^{u\dagger} c_{\phi q}^{(3)} y^u c_{\phi u}] - \mathrm{Tr}[y^{u\dagger} c_{\phi q}^{(3)} c_{\phi q}^{(1)} y^u] + \mathrm{Tr}[y^{u\dagger} c_{\phi q}^{(3)} c_{\phi q}^{(3)} y^u]) \Big) \Big\} \lambda$$

$$- \frac{2}{3} c_{\phi D} g_2^2 (c_{\phi\Box} + 4 \mathrm{Tr}[c_{\phi l}^{(3)}] + 12 \mathrm{Tr}[c_{\phi q}^{(3)}]) \lambda - \frac{1}{3} c_{\phi D} g_1^2 \Big( c_{\phi D} + c_{\phi\Box} - 4 \mathrm{Tr}[c_{\phi d}] - 4 \mathrm{Tr}[c_{\phi e}]$$

$$- 4 \mathrm{Tr}[c_{\phi l}^{(1)}] + 4 \mathrm{Tr}[c_{\phi q}^{(1)}] + 8 \mathrm{Tr}[c_{\phi u}] \Big) \lambda - \frac{3}{8} c_{\phi D}^2 (g_1^2 + g_2^2)^2 + 6 \Big( \mathrm{Tr}[y^d (c_{d\phi})^\dagger y^d (c_{d\phi})^\dagger]$$

$$+ 2 \mathrm{Tr}[y^d (c_{d\phi})^\dagger c_{d\phi} y^{d\dagger}] + \mathrm{Tr}[y^{d\dagger} c_{d\phi} y^{d\dagger} c_{d\phi}] + 2 \mathrm{Tr}[y^{d\dagger} c_{d\phi} (c_{d\phi})^\dagger y^d] \Big) + 2 \mathrm{Tr}[y^e (c_{e\phi})^\dagger y^e (c_{e\phi})^\dagger]$$

$$+ 4 \mathrm{Tr}[y^e (c_{e\phi})^\dagger c_{e\phi} y^{e\dagger}] + 2 \mathrm{Tr}[y^{e\dagger} c_{e\phi} y^{e\dagger} c_{e\phi}] + 4 \mathrm{Tr}[y^{e\dagger} c_{e\phi} (c_{e\phi})^\dagger y^e] + 6 \Big( \mathrm{Tr}[y^u (c_{u\phi})^\dagger y^u (c_{u\phi})^\dagger]$$

$$+ 2 \mathrm{Tr}[y^u (c_{u\phi})^\dagger c_{u\phi} y^{u\dagger}] + \mathrm{Tr}[y^{u\dagger} c_{u\phi} y^{u\dagger} c_{u\phi}] + 2 \mathrm{Tr}[y^{u\dagger} c_{u\phi} (c_{u\phi})^\dagger y^u] \Big)$$

$$- \frac{1}{3}\Big(3c_\phi + \lambda(c_{\phi D} - 8c_{\phi\Box})\Big)\Big(9c_{\phi D}g_1^2 - 9c_{\phi D}g_2^2 + 20c_{\phi\Box}g_2^2 + 12\lambda(c_{\phi D} - 2c_{\phi\Box})$$

$$+ 18\text{Tr}[y^d(c_{d\phi})^\dagger] + 18\text{Tr}[y^{d\dagger}c_{d\phi}] + 6\text{Tr}[y^e(c_{e\phi})^\dagger] + 6\text{Tr}[y^{e\dagger}c_{e\phi}] + 18\text{Tr}[y^u(c_{u\phi})^\dagger]$$

$$+ 18\text{Tr}[y^{u\dagger}c_{u\phi}] + 36\text{Tr}[y^d(c_{\phi ud})^\dagger y^{u\dagger}] - 72\text{Tr}[y^{d\dagger}c_{\phi q}^{(3)}y^d] - 24\text{Tr}[y^{e\dagger}c_{\phi l}^{(3)}y^e] + 36\text{Tr}[y^u c_{\phi ud}y^{d\dagger}]$$

$$- 72\text{Tr}[y^{u\dagger}c_{\phi q}^{(3)}y^u] + 8g_2^2(\text{Tr}[c_{\phi l}^{(3)}] + 3\text{Tr}[c_{\phi q}^{(3)}])\Big),$$

$$\dot{c}_{\phi^6}^{(1)} = -12c_\phi(c_{\phi D} + 8c_{\phi\Box}) + \frac{1}{24}(3g_1^2 + 157g_2^2 - 552\lambda)c_{\phi D}^2 - 3\text{Tr}[y^d(c_{d\phi})^\dagger]c_{\phi D} \qquad (83)$$

$$- 3\text{Tr}[y^{d\dagger}c_{d\phi}]c_{\phi D} - \text{Tr}[y^e(c_{e\phi})^\dagger]c_{\phi D} - \text{Tr}[y^{e\dagger}c_{e\phi}]c_{\phi D} - 3\text{Tr}[y^u(c_{u\phi})^\dagger]c_{\phi D}$$

$$- 3\text{Tr}[y^{u\dagger}c_{u\phi}]c_{\phi D} - 6\text{Tr}[y^d(c_{\phi ud})^\dagger y^{u\dagger}]c_{\phi D} + 12\text{Tr}[y^{d\dagger}c_{\phi q}^{(3)}y^d]c_{\phi D} + 4\text{Tr}[y^{e\dagger}c_{\phi l}^{(3)}y^e]c_{\phi D}$$

$$- 6\text{Tr}[y^u c_{\phi ud}y^{d\dagger}]c_{\phi D} - 2g_1^2\text{Tr}[c_{\phi d}]c_{\phi D} - 2g_1^2\text{Tr}[c_{\phi e}]c_{\phi D} - 2g_1^2\text{Tr}[c_{\phi l}^{(1)}]c_{\phi D} + \frac{8}{3}g_2^2\text{Tr}[c_{\phi l}^{(3)}]c_{\phi D}$$

$$+ 2g_1^2\text{Tr}[c_{\phi q}^{(1)}]c_{\phi D} + 8g_2^2\text{Tr}[c_{\phi q}^{(3)}]c_{\phi D} + 4g_1^2\text{Tr}[c_{\phi u}]c_{\phi D} + \Big(-2g_1^2 + \frac{29g_2^2}{2} - 64\lambda\Big)c_{\phi\Box}c_{\phi D}$$

$$+ \frac{4}{3}c_{\phi\Box}^2(15g_1^2 + 5g_2^2 + 264\lambda) - 24c_{\phi\Box}\text{Tr}[y^d(c_{d\phi})^\dagger] - 24c_{\phi\Box}\text{Tr}[y^{d\dagger}c_{d\phi}] - 8c_{\phi\Box}\text{Tr}[y^e(c_{e\phi})^\dagger]$$

$$- 8c_{\phi\Box}\text{Tr}[y^{e\dagger}c_{e\phi}] - 24c_{\phi\Box}\text{Tr}[y^u(c_{u\phi})^\dagger] - 24c_{\phi\Box}\text{Tr}[y^{u\dagger}c_{u\phi}] + 6\text{Tr}[(c_{d\phi})^\dagger c_{d\phi}] + 2\text{Tr}[(c_{e\phi})^\dagger c_{e\phi}]$$

$$+ 6\text{Tr}[(c_{u\phi})^\dagger c_{u\phi}] + 12g_2^2\text{Tr}[(c_{\phi d})^\dagger c_{\phi d}] + 4g_1^2\text{Tr}[(c_{\phi e})^\dagger c_{\phi e}] + 8g_1^2\text{Tr}[(c_{\phi l}^{(1)})^\dagger c_{\phi l}^{(1)}] + 8g_1^2\text{Tr}[(c_{\phi l}^{(3)})^\dagger c_{\phi l}^{(3)}]$$

$$- \frac{20}{3}g_2^2\text{Tr}[(c_{\phi l}^{(3)})^\dagger c_{\phi l}^{(3)}] + 24g_2^2\text{Tr}[(c_{\phi q}^{(1)})^\dagger c_{\phi q}^{(1)}] + 24g_1^2\text{Tr}[(c_{\phi q}^{(3)})^\dagger c_{\phi q}^{(3)}] - 20g_2^2\text{Tr}[(c_{\phi q}^{(3)})^\dagger c_{\phi q}^{(3)}]$$

$$+ 12g_2^2\text{Tr}[(c_{\phi u})^\dagger c_{\phi u}] - 3g_1^2\text{Tr}[(c_{\phi ud})^\dagger c_{\phi ud}] + 4g_2^2\text{Tr}[(c_{\phi ud})^\dagger c_{\phi ud}] + 18\text{Tr}[y^d(c_{d\phi})^\dagger c_{\phi q}^{(1)}]$$

$$+ 30\text{Tr}[y^d(c_{d\phi})^\dagger c_{\phi q}^{(3)}] - 48c_{\phi\Box}\text{Tr}[y^d(c_{\phi ud})^\dagger y^{u\dagger}] - 6\text{Tr}[y^d(c_{\phi ud})^\dagger(c_{u\phi})^\dagger] - 18\text{Tr}[y^{d\dagger}c_{d\phi}c_{\phi d}]$$

$$+ 96c_{\phi\Box}\text{Tr}[y^{d\dagger}c_{\phi q}^{(3)}y^d] + 6\text{Tr}[y^e(c_{e\phi})^\dagger c_{\phi l}^{(1)}] + 10\text{Tr}[y^e(c_{e\phi})^\dagger c_{\phi l}^{(3)}] - 6\text{Tr}[y^{e\dagger}c_{e\phi}c_{\phi e}]$$

$$+ 32c_{\phi\Box}\text{Tr}[y^{e\dagger}c_{\phi l}^{(3)}y^e] + 18\text{Tr}[y^u c_{\phi u}(c_{u\phi})^\dagger] - 48c_{\phi\Box}\text{Tr}[y^u c_{\phi ud}y^{d\dagger}] - 6\text{Tr}[y^{u\dagger}c_{d\phi}(c_{\phi ud})^\dagger]$$

$$- 18\text{Tr}[y^{u\dagger}c_{\phi q}^{(1)}c_{u\phi}] + 12(c_{\phi D} + 8c_{\phi\Box})\text{Tr}[y^{u\dagger}c_{\phi q}^{(3)}y^u] + 30\text{Tr}[y^{u\dagger}c_{\phi q}^{(3)}c_{u\phi}] + 18\text{Tr}[c_{u\phi}c_{\phi u}y^{u\dagger}]$$

$$- 6\text{Tr}[c_{u\phi}c_{\phi ud}y^{d\dagger}] - 18\text{Tr}[c_{\phi d}(c_{d\phi})^\dagger y^d] - 6\text{Tr}[c_{\phi e}(c_{e\phi})^\dagger y^e] + 6\text{Tr}[c_{\phi l}^{(1)}c_{e\phi}y^{e\dagger}] + 10\text{Tr}[c_{\phi l}^{(3)}c_{e\phi}y^{e\dagger}]$$

$$+ 18\text{Tr}[c_{\phi q}^{(1)}c_{d\phi}y^{d\dagger}] + 30\text{Tr}[c_{\phi q}^{(3)}c_{d\phi}y^{d\dagger}] - 6\text{Tr}[c_{\phi ud}(c_{d\phi})^\dagger y^u] - 18\text{Tr}[(c_{u\phi})^\dagger c_{\phi q}^{(1)}y^u]$$

$$+ 30\text{Tr}[(c_{u\phi})^\dagger c_{\phi q}^{(3)}y^u] + 36\text{Tr}[y^d c_{\phi d}y^{d\dagger}c_{\phi q}^{(1)}] + 36\text{Tr}[y^d c_{\phi d}y^{d\dagger}c_{\phi q}^{(3)}] - 18\text{Tr}[y^d c_{\phi d}c_{\phi d}y^{d\dagger}]$$

$$- 3\text{Tr}[y^d(c_{\phi ud})^\dagger c_{\phi ud}y^{d\dagger}] - 18\text{Tr}[y^{d\dagger}c_{\phi q}^{(1)}c_{\phi q}^{(1)}y^d] - 18\text{Tr}[y^{d\dagger}c_{\phi q}^{(1)}c_{\phi q}^{(3)}y^d] + 12\text{Tr}[y^{d\dagger}c_{\phi q}^{(3)}y^u c_{\phi ud}]$$

$$- 18\text{Tr}[y^{d\dagger}c_{\phi q}^{(3)}c_{\phi q}^{(1)}y^d] - 30\text{Tr}[y^{d\dagger}c_{\phi q}^{(3)}c_{\phi q}^{(3)}y^d] + 12\text{Tr}[y^e c_{\phi e}y^{e\dagger}c_{\phi l}^{(1)}] + 12\text{Tr}[y^e c_{\phi e}y^{e\dagger}c_{\phi l}^{(3)}]$$

$$- 6\text{Tr}[y^e c_{\phi e}c_{\phi e}y^{e\dagger}] - 6\text{Tr}[y^{e\dagger}c_{\phi l}^{(1)}c_{\phi l}^{(1)}y^e] - 6\text{Tr}[y^{e\dagger}c_{\phi l}^{(1)}c_{\phi l}^{(3)}y^e] - 6\text{Tr}[y^{e\dagger}c_{\phi l}^{(3)}c_{\phi l}^{(1)}y^e]$$

$$- 10\text{Tr}[y^{e\dagger}c_{\phi l}^{(3)}c_{\phi l}^{(3)}y^e] - 18\text{Tr}[y^u c_{\phi u}c_{\phi u}y^{u\dagger}] - 3\text{Tr}[y^u c_{\phi ud}(c_{\phi ud})^\dagger y^{u\dagger}] + 36\text{Tr}[y^{u\dagger}c_{\phi q}^{(1)}y^u c_{\phi u}]$$

$$- 18\text{Tr}[y^{u\dagger}c_{\phi q}^{(1)}c_{\phi q}^{(1)}y^u] + 18\text{Tr}[y^{u\dagger}c_{\phi q}^{(1)}c_{\phi q}^{(3)}y^u] + 12\text{Tr}[y^{u\dagger}c_{\phi q}^{(3)}y^d(c_{\phi ud})^\dagger] - 36\text{Tr}[y^{u\dagger}c_{\phi q}^{(3)}y^u c_{\phi u}]$$

$$+ 18\text{Tr}[y^{u\dagger}c_{\phi q}^{(3)}c_{\phi q}^{(1)}y^u] - 30\text{Tr}[y^{u\dagger}c_{\phi q}^{(3)}c_{\phi q}^{(3)}y^u] - \frac{32}{3}c_{\phi\Box}g_2^2\text{Tr}[c_{\phi l}^{(3)}] - 32c_{\phi\Box}g_2^2\text{Tr}[c_{\phi q}^{(3)}],$$

$$\dot{c}_{\phi^6}^{(2)} = -\frac{2}{3}\Big(3\text{Tr}[(c_{\phi d})^\dagger c_{\phi d}] + \text{Tr}[(c_{\phi e})^\dagger c_{\phi e}] + 2\text{Tr}[(c_{\phi l}^{(1)})^\dagger c_{\phi l}^{(1)}] + 6\text{Tr}[(c_{\phi q}^{(1)})^\dagger c_{\phi q}^{(1)}]\Big)g_2^2 \quad (84)$$

$$- 18c_\phi c_{\phi D} + \frac{1}{12}\Big((-35g_1^2 + 46g_2^2 - 312\lambda)c_{\phi D}^2 + 4c_{\phi\Box}(20g_1^2 + 15g_2^2 + 192\lambda)c_{\phi D} + 160c_{\phi\Box}^2 g_1^2\Big)$$

$$+ 12\text{Tr}[y^d(c_{d\phi})^\dagger c_{\phi q}^{(1)}] + 6\text{Tr}[y^d(c_{\phi ud})^\dagger(c_{u\phi})^\dagger] - 12\text{Tr}[y^{d\dagger}c_{d\phi}c_{\phi d}] + 4\text{Tr}[y^e(c_{e\phi})^\dagger c_{\phi l}^{(1)}]$$

$$- 4\text{Tr}[y^{e\dagger}c_{e\phi}c_{\phi e}] + 12\text{Tr}[y^u c_{\phi u}(c_{u\phi})^\dagger] + 6\text{Tr}[y^{u\dagger}c_{d\phi}(c_{\phi ud})^\dagger] - 12\text{Tr}[y^{u\dagger}c_{\phi q}^{(1)}c_{u\phi}]$$

$$+ 12\text{Tr}[c_{u\phi}c_{\phi u}y^{u\dagger}] + 6\text{Tr}[c_{u\phi}c_{\phi ud}y^{d\dagger}] - 12\text{Tr}[c_{\phi d}(c_{d\phi})^\dagger y^d] - 4\text{Tr}[c_{\phi e}(c_{e\phi})^\dagger y^e] + 4\text{Tr}[c_{\phi l}^{(1)}c_{e\phi}y^{e\dagger}]$$

$$+ 12\text{Tr}[c_{\phi q}^{(1)}c_{d\phi}y^{d\dagger}] + 6\text{Tr}[c_{\phi ud}(c_{d\phi})^\dagger y^u] - 12\text{Tr}[(c_{u\phi})^\dagger c_{\phi q}^{(1)}y^u] + 24\text{Tr}[y^d c_{\phi d}y^{d\dagger}c_{\phi q}^{(1)}]$$

$$+ 24\text{Tr}[y^d c_{\phi d}y^{d\dagger}c_{\phi q}^{(3)}] - 12\text{Tr}[y^d c_{\phi d}c_{\phi d}y^{d\dagger}] + 3\text{Tr}[y^d(c_{\phi ud})^\dagger c_{\phi ud}y^{d\dagger}] - 12\text{Tr}[y^{d\dagger}c_{\phi q}^{(1)}c_{\phi q}^{(1)}y^d]$$

$$- 12\text{Tr}[y^{d\dagger}c_{\phi q}^{(1)}c_{\phi q}^{(3)}y^d] - 12\text{Tr}[y^{d\dagger}c_{\phi q}^{(3)}y^u c_{\phi ud}] - 12\text{Tr}[y^{d\dagger}c_{\phi q}^{(3)}c_{\phi q}^{(1)}y^d] + 8\text{Tr}[y^e c_{\phi e}y^{e\dagger}c_{\phi l}^{(1)}]$$

$$+ 8\text{Tr}[y^e c_{\phi e}y^{e\dagger}c_{\phi l}^{(3)}] - 4\text{Tr}[y^e c_{\phi e}c_{\phi e}y^{e\dagger}] - 4\text{Tr}[y^{e\dagger}c_{\phi l}^{(1)}c_{\phi l}^{(1)}y^e] - 4\text{Tr}[y^{e\dagger}c_{\phi l}^{(1)}c_{\phi l}^{(3)}y^e]$$

$$- 4\text{Tr}[y^{e\dagger}c_{\phi l}^{(3)}c_{\phi l}^{(1)}y^e] - 12\text{Tr}[y^u c_{\phi u}c_{\phi u}y^{u\dagger}] + 3\text{Tr}[y^u c_{\phi ud}(c_{\phi ud})^\dagger y^{u\dagger}] + 24\text{Tr}[y^{u\dagger}c_{\phi q}^{(1)}y^u c_{\phi u}]$$

$$- 12\text{Tr}[y^{u\dagger}c_{\phi q}^{(1)}c_{\phi q}^{(1)}y^u] + 12\text{Tr}[y^{u\dagger}c_{\phi q}^{(1)}c_{\phi q}^{(3)}y^u] - 12\text{Tr}[y^{u\dagger}c_{\phi q}^{(3)}y^d(c_{\phi ud})^\dagger] - 24\text{Tr}[y^{u\dagger}c_{\phi q}^{(3)}y^u c_{\phi u}]$$

$$+ 12\text{Tr}[y^{u\dagger}c_{\phi q}^{(3)}c_{\phi q}^{(1)}y^u] + \frac{1}{3}\bigg(16\text{Tr}[(c_{\phi l}^{(3)})^\dagger c_{\phi l}^{(3)}]g_1^2 + 48\text{Tr}[(c_{\phi q}^{(3)})^\dagger c_{\phi q}^{(3)}]g_1^2 - 6\text{Tr}[(c_{\phi ud})^\dagger c_{\phi ud}]g_1^2$$

$$- 4c_{\phi D}\text{Tr}[c_{\phi d}]g_1^2 - 4c_{\phi D}\text{Tr}[c_{\phi e}]g_1^2 - 4c_{\phi D}\text{Tr}[c_{\phi l}^{(1)}]g_1^2 + 4c_{\phi D}\text{Tr}[c_{\phi q}^{(1)}]g_1^2 + 8c_{\phi D}\text{Tr}[c_{\phi u}]g_1^2$$

$$- 18c_{\phi D}\text{Tr}[y^d(c_{d\phi})^\dagger] - 18c_{\phi D}\text{Tr}[y^{d\dagger}c_{d\phi}] - 6c_{\phi D}\text{Tr}[y^e(c_{e\phi})^\dagger] - 6c_{\phi D}\text{Tr}[y^{e\dagger}c_{e\phi}]$$

$$- 18c_{\phi D}\text{Tr}[y^u(c_{u\phi})^\dagger] - 18c_{\phi D}\text{Tr}[y^{u\dagger}c_{u\phi}] - 6g_2^2\text{Tr}[(c_{\phi u})^\dagger c_{\phi u}] + 3g_2^2\text{Tr}[(c_{\phi ud})^\dagger c_{\phi ud}]$$

$$- 36c_{\phi D}\text{Tr}[y^d(c_{\phi ud})^\dagger y^{u\dagger}] + 72c_{\phi D}\text{Tr}[y^{d\dagger}c_{\phi q}^{(3)}y^d] + 24c_{\phi D}\text{Tr}[y^{e\dagger}c_{\phi l}^{(3)}y^e] - 36c_{\phi D}\text{Tr}[y^u c_{\phi ud}y^{d\dagger}]$$

$$+ 72c_{\phi D}\text{Tr}[y^{u\dagger}c_{\phi q}^{(3)}y^u]\bigg),$$

$$\dot{c}_{\phi^4}^{(1)} = \frac{1}{3}\bigg(24\text{Tr}[(c_{\phi d})^\dagger c_{\phi d}] + 8\text{Tr}[(c_{\phi e})^\dagger c_{\phi e}] + 16\text{Tr}[(c_{\phi l}^{(1)})^\dagger c_{\phi l}^{(1)}] - 16\text{Tr}[(c_{\phi l}^{(3)})^\dagger c_{\phi l}^{(3)}] \quad (85)$$

$$+ 48\text{Tr}[(c_{\phi q}^{(1)})^\dagger c_{\phi q}^{(1)}] - 48\text{Tr}[(c_{\phi q}^{(3)})^\dagger c_{\phi q}^{(3)}] + 24\text{Tr}[(c_{\phi u})^\dagger c_{\phi u}] - 24\text{Tr}[(c_{\phi ud})^\dagger c_{\phi ud}] + 32c_{\phi D}c_{\phi\Box}$$

$$- 11c_{\phi D}^2 - 16c_{\phi\Box}^2\bigg),$$

$$\dot{c}_{\phi^4}^{(2)} = \frac{1}{3}\bigg(- 8(3\text{Tr}[(c_{\phi d})^\dagger c_{\phi d}] + \text{Tr}[(c_{\phi e})^\dagger c_{\phi e}] + 2\text{Tr}[(c_{\phi l}^{(1)})^\dagger c_{\phi l}^{(1)}] + 2\text{Tr}[(c_{\phi l}^{(3)})^\dagger c_{\phi l}^{(3)}] \quad (86)$$

$$+ 6\text{Tr}[(c_{\phi q}^{(1)})^\dagger c_{\phi q}^{(1)}] + 6\text{Tr}[(c_{\phi q}^{(3)})^\dagger c_{\phi q}^{(3)}] + 3\text{Tr}[(c_{\phi u})^\dagger c_{\phi u}]) - 16c_{\phi D}c_{\phi\Box} - 5c_{\phi D}^2 - 16c_{\phi\Box}^2\bigg),$$

$$\dot{c}_{\phi^4}^{(3)} = \frac{1}{3}\bigg(32\text{Tr}[(c_{\phi l}^{(3)})^\dagger c_{\phi l}^{(3)}] + 96\text{Tr}[(c_{\phi q}^{(3)})^\dagger c_{\phi q}^{(3)}] + 24\text{Tr}[(c_{\phi ud})^\dagger c_{\phi ud}] - 16c_{\phi D}c_{\phi\Box} \quad (87)$$

$$+ 7c_{\phi D}^2 - 40c_{\phi\Box}^2\bigg),$$

$$\dot{c}_{W^2\phi^4}^{(1)} = -\frac{1}{24}g_2^2\bigg(3c_{\phi D}^2 + 24\text{Tr}[(c_{\phi d})^\dagger c_{\phi d}] - 48\text{Tr}[(c_{\phi l}^{(3)})^\dagger c_{\phi l}^{(3)}] + 48\text{Tr}[(c_{\phi q}^{(1)})^\dagger c_{\phi q}^{(1)}] \quad (88)$$

$$- 144\text{Tr}[(c_{\phi q}^{(3)})^\dagger c_{\phi q}^{(3)}] + 24\text{Tr}[(c_{\phi u})^\dagger c_{\phi u}] + 24\text{Tr}[(c_{\phi ud})^\dagger c_{\phi ud}] + 8\text{Tr}[(c_{\phi e})^\dagger c_{\phi e}]$$

$$+ 16\text{Tr}[(c_{\phi l}^{(1)})^\dagger c_{\phi l}^{(1)}]\bigg),$$

$$\dot{c}_{W\phi^4 D^2}^{(1)} = -\frac{1}{3}g_2\bigg(3c_{\phi D}^2 + 24\text{Tr}[(c_{\phi d})^\dagger c_{\phi d}] - 48\text{Tr}[(c_{\phi l}^{(3)})^\dagger c_{\phi l}^{(3)}] + 48\text{Tr}[(c_{\phi q}^{(1)})^\dagger c_{\phi q}^{(1)}] \quad (89)$$

$$- 144\text{Tr}[(c_{\phi q}^{(3)})^\dagger c_{\phi q}^{(3)}] + 24\text{Tr}[(c_{\phi u})^\dagger c_{\phi u}] + 24\text{Tr}[(c_{\phi ud})^\dagger c_{\phi ud}] + 8\text{Tr}[(c_{\phi e})^\dagger c_{\phi e}]$$

$$+ 16\text{Tr}[(c_{\phi l}^{(1)})^\dagger c_{\phi l}^{(1)}]\bigg),$$

$$\dot{c}^{(1)}_{B^2\phi^4} = \frac{1}{24}g_1^2\left(3c_{\phi D}^2 + 24\text{Tr}[(c_{\phi d})^\dagger c_{\phi d}] - 48\text{Tr}[(c_{\phi l}^{(3)})^\dagger c_{\phi l}^{(3)}] + 48\text{Tr}[(c_{\phi q}^{(1)})^\dagger c_{\phi q}^{(1)}]\right. \tag{90}$$

$$- 144\text{Tr}[(c_{\phi q}^{(3)})^\dagger c_{\phi q}^{(3)}] + 24\text{Tr}[(c_{\phi u})^\dagger c_{\phi u}] + 24\text{Tr}[(c_{\phi ud})^\dagger c_{\phi ud}] + 8\text{Tr}[(c_{\phi e})^\dagger c_{\phi e}]$$

$$\left.+ 16\text{Tr}[(c_{\phi l}^{(1)})^\dagger c_{\phi l}^{(1)}]\right),$$

$$\dot{c}^{(1)}_{B\phi^4 D^2} = \frac{1}{3}g_1\left(3c_{\phi D}^2 + 24\text{Tr}[(c_{\phi d})^\dagger c_{\phi d}] - 48\text{Tr}[(c_{\phi l}^{(3)})^\dagger c_{\phi l}^{(3)}] + 48\text{Tr}[(c_{\phi q}^{(1)})^\dagger c_{\phi q}^{(1)}]\right. \tag{91}$$

$$- 144\text{Tr}[(c_{\phi q}^{(3)})^\dagger c_{\phi q}^{(3)}] + 24\text{Tr}[(c_{\phi u})^\dagger c_{\phi u}] + 24\text{Tr}[(c_{\phi ud})^\dagger c_{\phi ud}] + 8\text{Tr}[(c_{\phi e})^\dagger c_{\phi e}]$$

$$\left.+ 16\text{Tr}[(c_{\phi l}^{(1)})^\dagger c_{\phi l}^{(1)}]\right).$$

## C   Ultraviolet completion of the Standard Model

The purpose of this appendix is proving that there exists at least one UV completion of the SM that induces arbitrary values of $c_\phi$, $c_{\phi D}$ and $c_{\phi\square}$. To this aim, let us extend the SM (for $\mu^2 = 0$) with three colorless scalars: $\mathcal{S} \sim (1,1)_0$, $\Xi_0 \sim (1,3)_0$ and $\Xi_1 \sim (1,3)_1$. The numbers in parentheses and the subscript indicate the representations of $SU(3)_c$, $SU(2)_L$ and $U(1)_Y$, respectively.

Let us assume that they all have mass $M$ much larger than the EW scale, and that the new physics interaction Lagrangian is:

$$\mathcal{L}_{\text{NP}} = \kappa_\mathcal{S}\mathcal{S}\phi^\dagger\phi + \lambda_\mathcal{S}\mathcal{S}^2\phi^\dagger\phi + \kappa_{\Xi_0}\phi^\dagger\Xi_0^a\sigma_a\phi + \left(\kappa_{\Xi_1}\Xi_1^{a\dagger}\tilde{\phi}^\dagger\sigma_a\phi + \text{h.c.}\right). \tag{92}$$

(Other triple and quartic terms are allowed, but we just ignored them for simplicity.) Then, by integrating out the heavy modes at tree level at the scale $M$, we obtain [46]:

$$\frac{c_\phi}{\Lambda^2} = -\lambda_\mathcal{S}\frac{\kappa_\mathcal{S}^2}{M^4},$$

$$\frac{c_{\phi D}}{\Lambda^2} = \frac{2}{M^4}(2\kappa_{\Xi_1}^2 - \kappa_{\Xi_0}^2),$$

$$\frac{c_{\phi\square}}{\Lambda^2} = \frac{1}{2M^4}(4\kappa_{\Xi_1}^2 + \kappa_{\Xi_0}^2 - \kappa_\mathcal{S}^2). \tag{93}$$

Obviously, $c_\phi$ can have arbitrary sign by just tuning $\lambda_\mathcal{S}$. Likewise, $c_{\phi D}$ can be made arbitrarily negative provided $\kappa_{\Xi_1}/\kappa_{\Xi_0} \ll 1$, and positive otherwise. Notwithstanding this later choice, $c_{\phi\square}$ will be positive for small enough $\kappa_\mathcal{S}$ and negative for large values of this parameter. In summary, the signs of the three tree-level generated dimension-six operators are arbitrary and uncorrelated.

In the process of integrating out the fields of mass $M$, dimension-eight operators arise too. With the help of `MatchingTools` [74], we find that (see also Ref. [33]):

$$\frac{c_{\phi^4}^{(1)}}{\Lambda^4} = 4\frac{\kappa_{\Xi_0}^2}{M^6}, \qquad \frac{c_{\phi^4}^{(2)}}{\Lambda^4} = 8\frac{\kappa_{\Xi_1}^2}{M^6}, \qquad \frac{c_{\phi^4}^{(3)}}{\Lambda^4} = \frac{2}{M^6}(\kappa_\mathcal{S}^2 - \kappa_{\Xi_0}^2). \tag{94}$$

Contrary to the dimension-six Wilson coefficients above, these couplings fullfill the positivity bounds $c_{\phi^4}^{(2)} \geq 0$, $c_{\phi^4}^{(1)} + c_{\phi^4}^{(2)} \geq 0$ and $c_{\phi^4}^{(1)} + c_{\phi^4}^{(2)} + c_{\phi^4}^{(3)} \geq 0$ obtained in Ref. [33] for arbitrary values of the $\kappa$s.

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
