# Peer review of "Towards the renormalisation of the Standard Model effective field theory to dimension eight: Bosonic interactions I"

_SciPost Physics_

## Round 1 · Referee Report · Anonymous (Referee 1) · 2021-7-5

Report

In this paper, the contributions of pairs of dimension-6 operators in the SMEFT to the renormalization group evolution of dimension-8 operators are calculated. Some consequences of the results are also briefly explored, including their interplay with positivity bounds and their effects in constraints from experimental data. It is thus relevant to the current developments in the SMEFT, and a first step towards the full calculation of renormalization group running of dimension-8 operators. As such, it meets the criteria for publication in SciPost Physics, and I recommend publication after the following minor points have been addressed:

  • Tables 2 and 3 do not show redundant bosonic operators not containing the Higgs, such as (D^mu B_munu)^2. Some justification for this would be welcome.

  • In eq. (1): the sign of the negative mass term for the Higgs doublet is reversed.

  • The links to arXiv in the references lead to non-existent SciPost pages. This happens in the SciPost version of the paper but not in the arXiv one.

  • validity: -
  • significance: -
  • originality: -
  • clarity: -
  • formatting: -
  • grammar: -

Author:  Maria Ramos  on 2021-07-06  [id 1548]

(in reply to Report 1 on 2021-07-05)

We would like to thank the referee for the positive report.

Our replies to the comments raised by the referee are listed below:

1) We did not build the redundant bosonic operators not involving the Higgs because they are not renormalised by the interactions in Eq. (4). We could have constructed them for completeness, but that is a highly non-trivial task that we will addressed in a forecoming publication. That is the reason why we did not include them in the tables. In order to clarify this point, we will add a comment in this respect in the future version of the manuscript.

2) The sign of the Higgs mass term in the Lagrangian is a convention. Still, we think changing our choice of sign is a good idea in order to match the convention adopted in the papers addressing renormalisation to dimension six [1312.2014, 1310.4838, 1308.2627].

3) We will try to fix the arXiv links in the references in the future version of the manuscript.

Thank you very much.

Best wishes, The authors.

---

## Editorial Decision

resubmitted